# Neural encoding of actual and imagined touch within human posterior parietal cortex

Srinivas Chivukula[1,2,3†], Carey Y Zhang[1,2†], Tyson Aflalo[1,2†*], Matiar Jafari[1,2,3], Kelsie Pejsa[1,2], Nader Pouratian[1,2,3], Richard A Andersen[1,2]

[1]Department of Biology and Biological Engineering, California Institute of Technology, Pasadena, United States; [2]Tianqiao and Chrissy Chen Brain-Machine Interface Center, Chen Institute for Neuroscience, California Institute of Technology, Pasadena, United States; [3]Geffen School of Medicine, University of California, Los Angeles, Los Angeles, United States

**Abstract** In the human posterior parietal cortex (PPC), single units encode high-dimensional information with *partially mixed* representations that enable small populations of neurons to encode many variables relevant to movement planning, execution, cognition, and perception. Here, we test whether a PPC neuronal population previously demonstrated to encode visual and motor information is similarly engaged in the somatosensory domain. We recorded neurons within the PPC of a human clinical trial participant during actual touch presentation and during a tactile imagery task. Neurons encoded actual touch at short latency with bilateral receptive fields, organized by body part, and covered all tested regions. The tactile imagery task evoked body part-specific responses that shared a neural substrate with actual touch. Our results are the first neuron-level evidence of touch encoding in human PPC and its cognitive engagement during a tactile imagery task, which may reflect semantic processing, attention, sensory anticipation, or imagined touch.

**\*For correspondence:**
taflalo@caltech.edu

[†]These authors contributed equally to this work

**Competing interests:** The authors declare that no competing interests exist.

## Introduction

Touch is a complex, multisensory perceptual process (*de Haan and Dijkerman, 2020*; *de Lafuente and Romo, 2006*; *Graziano and Gross, 1993*). In non-human primates (NHPs), multisensory input (e.g., visual, tactile) converges upon neurons in higher-order brain regions such as the posterior parietal cortex (PPC) where they are integrated into coherent representations (*Graziano and Gross, 1993*; *Avillac et al., 2007*; *Graziano, 1999*; *Graziano, 2001*; *Graziano et al., 2000*; *Holmes and Spence, 2004*; *Hwang et al., 2014*; *Seelke et al., 2012*; *Sereno and Huang, 2014*). Recent human neuroimaging studies suggest that the PPC is also recruited during touch cognition in the absence of actual tactile input (e.g., seen touch or imagined touch), supporting a notion that both higher-level touch processing and tactile cognition share a neural substrate (*Chan and Baker, 2015*; *Lucas et al., 2015*). To date, however, such a link has not been established at the single neuron level.

We recently reported an analogous relation in the parallel domain of motor function (*Aflalo et al., 2020*; *Aflalo et al., 2015*; *Rutishauser et al., 2018*; *Zhang et al., 2017*). In these studies, we found that a shared PPC neuronal population coded for overt movements as well as cognitive motor variables including imagery, observed actions, and action verbs (*Aflalo et al., 2020*; *Aflalo et al., 2015*; *Andersen and Buneo, 2002*; *Rutishauser et al., 2018*; *Zhang et al., 2017*). This richness of representation is made possible through a *partially mixed* encoding in which single neurons represent multiple variables, allowing a relatively small neuronal population (recorded through

a 4 × 4 mm implanted microelectrode array) to provide many movement-related signals (*Zhang et al., 2017*; *Zhang et al., 2020*). Here, we hypothesize that the same PPC neuronal population engaged by high-level motor cognition also encodes actual tactile sensations as well as tactile cognition within this partially mixed encoding structure.

The neural correlates of somatosensory perception are characterized by spatially structured receptive fields to touch that respond at short latency (*Keysers et al., 2010*). In NHPs, subregions of the PPC within and medial to the intraparietal sulcus (IPS) encode tactile receptive fields that respond to bilateral stimuli (*Graziano and Gross, 1993*; *Avillac et al., 2007*; *Graziano, 1999*; *Seelke et al., 2012*; *Sereno and Huang, 2014*). These are often large receptive fields, extending across multiple body parts (*Avillac et al., 2007*; *Graziano, 1999*; *Graziano, 2001*; *Graziano et al., 2000*; *Hwang et al., 2014*; *Seelke et al., 2012*; *Sereno and Huang, 2014*; *Sakata et al., 1973*). In humans, functional magnetic resonance imaging (fMRI) studies support multisensory encoding of touch within and medial to the IPS in anterior portions of PPC (*Huang et al., 2018*; *Sereno and Huang, 2014*; *Huang et al., 2012*). Although these studies indicate that relatively small regions of PPC may encode touch to large portions of the body, the limited spatial resolution of fMRI precludes a characterization of tactile receptive fields. The inability to resolve single neurons in fMRI is especially problematic when attempting to understand the significance of the grossly overlapping representations of actual touch and cognitive representations of touch (*Chan and Baker, 2015*; *Lucas et al., 2015*). Spatial correspondence in fMRI cannot confirm whether representations share a neuron-level substrate (*Caramazza et al., 2014*). Taken together, it is unclear from the current literature whether individual neurons in human PPC discriminate touch to different segments of the body with spatially structured receptive fields, and, if so, whether cognitive processing of touch engages the same populations of cells.

In a unique opportunity, we investigated touch processing in a tetraplegic human subject at the level of single neurons recorded from an electrode array implanted in the left PPC for an ongoing brain machine interface (BMI) clinical trial. We recorded single- and multi-unit neural activity during the presentation of actual touch and during imagined touch to sensate dermatomes above the level of the participant's injury. We found that neurons recorded at the junction of the postcental and intraparietal sulci in humans (postcentral-intraparietal, PC-IP) encoded actual touch at short latency (~50 ms) with bilateral spatially structured receptive fields, covering all tested, sensate regions within the head, face, neck, and shoulders. The tactile imagery task evoked body part-specific responses that shared a neural substrate with actual touch. Our results demonstrate that PPC neurons that discriminate touch are partially reactivated during a tactile imagery task in a body part-specific manner. The latter represents a novel finding, thus far untestable in NHP models, and suggests PPC involvement in the cognitive processing of touch.

## Results

We recorded from on average 101.6 ± 7.2 neurons (*Figure 1—figure supplement 1*) over 14 sessions in the PPC (left hemisphere) of a tetraplegic human participant (spinal injury at levels 3–4; C3/4). In previous work, we referred to the implant area as the anterior intraparietal cortex, a region functionally defined in NHPs (*Aflalo et al., 2020*, *Aflalo et al., 2015*; *Rutishauser et al., 2018*; *Zhang et al., 2017*; *Zhang et al., 2020*; *Andersen et al., 2019*; *Sakellaridi et al., 2019*). Here, we refer to the recording site as the PC-IP, acknowledging that further work is necessary to definitively characterize homologies between human and NHP anatomy. Recordings were split across four tasks, designed to probe basic properties of the neuronal population during both actual and imagined touch. Recordings were made from a chronic implanted array, and thus neuronal waveform sorting resulted in both well-isolated neuronal waveforms and multi-neuron groupings. The main figures aggregate across sorted channels while key analyses are performed separately for well-isolated and multi-unit activity in supplemental figures.

### PC-IP neurons encode bilateral tactile receptive fields

We first examined the hypothesis that PC-IP neurons encode tactile receptive fields to dermatomes above the level of the participant's spinal cord injury (SCI). Tactile stimuli were delivered as rubbing motions at approximately 1 Hz for 3 s. The subject was asked to keep her eyes closed to eliminate neural responses arising from visual input. Tactile stimuli were presented to bilateral axial (forehead,

vertex, cheek, neck, back) and truncal (shoulder) body parts to determine the extent of body coverage of any tactile representations among PC-IP neurons. As controls, touch was also presented to the bilateral hands (insensate regions below the level of SCI), and a null condition was included (with no stimulus delivered) to verify that touch-related neural responses did not arise by chance.

For each neuron, we fit a linear model that explained firing rate as a function of responses to each touch location. Neural responses to a particular body location were considered significant if the t-statistic for the associated beta coefficient was significant (p<0.05, false discovery rate [FDR] corrected for multiple comparisons). A significant fraction of the neuronal population encoded touch to each of the tested body parts with preserved somatosensation ($\chi^2$(1)=3908.98, p<0.05; *Figure 1A*, *Figure 1—figure supplement 2*). These results are consistent with bilateral encoding as the tested body parts included both body sides. Neither touch to the hands nor the null condition elicited significant neuronal modulation. Single neurons discriminated the location of actual touch: Of the 263 responsive units shown in *Figure 1A*, we found that 257 discriminated touch location (ANOVA, FDR corrected for multiple comparisons). Representative examples of neurons showing clear discrimination between the different touch locations are shown in *Figure 1B*. As expected, a population of discriminative cells enabled accurate cross-validated classification of the touched body part (*Figure 1C*; see *Figure 1—figure supplement 3* for single-session examples).

Single neurons were heterogenous, responding to variable numbers of touch sites (*Figure 2A*, *Figure 2—figure supplement 1*). Right and left sides tended to respond to the same number of fields (evidenced by the strong diagonal structure of *Figure 2A*). Tactile receptive fields of PC-IP neurons were diverse with evidence both for broad single-peaked fields and multi-peaked fields characterized by spatially separated regions of enhanced response (*Figure 2—figure supplement 2*, *Figure 2—figure supplement 3*).

PC-IP neurons demonstrated mirror-symmetric bilateral coding. We performed a cross-validated population correlation analysis to measure population-level similarity in the responses to each touch location (*Figure 2B*, *Figure 2—figure supplement 4*). In brief, the neural activation pattern elicited by touch to each body location was quantified as a vector, with each vector element capturing the mean response for a particular neuron during actual touch. These vectors were then pairwise correlated in a cross-validated manner so that the strength of correlation between any two body parts could be compared against the strength of correlation for repeated touches applied to the same body part. We found that responses to the same touch locations on the right and left sides are highly correlated, comparable to the correlation for repeated touches applied to the same body part. This result is consistent with a strong, mirror-symmetric, bilateral encoding. As expected, correlations involving the hands and the null condition were distributed about zero, consistent with a lack of systematic neural population response to these conditions. The results from the correlation analysis were similar for alternative distance metrics (*Figure 2—figure supplement 5*). Further, analysis of single units revealed mirror symmetry in bilateral representation for the vast majority of the population, paralleling population-level findings (*Figure 2—figure supplement 6*).

## Tactile responses occur at short latency to bilateral stimuli

We explored PC-IP population response latency to tactile stimulation on the contralateral and ipsilateral body sides. In a variation of the basic task paradigm, we used a capacitive touch sensing probe to acquire precise measurements of the time of contact with the skin surface in order to measure the latency in neuronal response from the time of tactile stimulation. We probed latency on the bilateral cheeks and shoulders. As a control, we included both hands in the task design.

We measured latency as the time at which the response of the neural population rose above the pre-stimulus baseline activity (*Figure 3*). The neural population response was quantified as the first principal component computed from principal component analysis (PCA) of the activity of all neurons (*Cunningham and Yu, 2014*; *Cunningham and Ghahramani, 2015*). The first principal component was then fit with a piecewise linear function, and latency was computed as the time the linear function crossed the baseline pre-stimulus response. Response latency was short for both body sides and slightly shorter for contralateral (right) receptive fields (50 ms) than for ipsilateral (left) receptive fields (54 ms), although this difference was not statistically significant (permutation shuffle test, p>0.05). *Figure 3A* shows the time course of the first principal component relative to time of contact of the touch probe (stepped window; 2 ms window size, stepped at 2 ms, no smoothing) along with

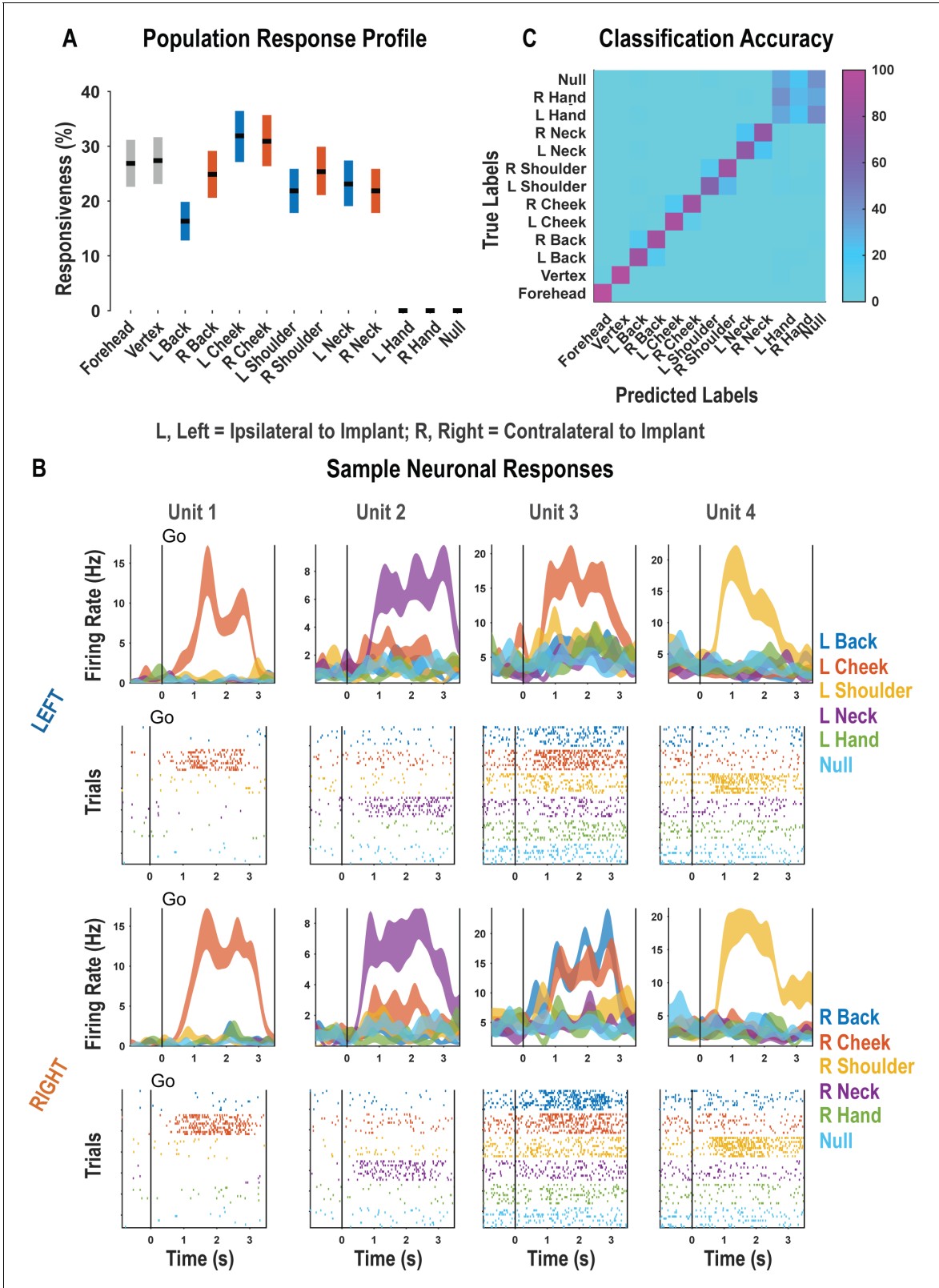

**Figure 1.** Postcentral-intraparietal (PC-IP) discriminably encodes bilateral tactile receptive fields. (**A**) Percentage of the PC-IP neuronal population that demonstrated significant modulation relative to baseline for each tested stimulation site (p<0.05, false discovery rate corrected, n = 398 units). Results are shown as the mean percentage (horizontal black line) of the population ± bootstrapped 95% confidence interval (bar height). Gray bars represent truncal (midline) body locations, blue bars represent left (ipsilateral)-sided sites, and orange bars represent right (contralateral)-sided sites. Population

*Figure 1 continued on next page*

*Figure 1 continued*

results were pooled across recording sessions (*Figure 1—figure supplement 1*) and were not qualitatively affected by pooling together single and potential multi-units (*Figure 1—figure supplement 2*). (B) Representative neuronal responses illustrating body part discrimination. Each column of panels depicts the response for one neuron to body parts on the left (top two rows) and on the right (bottom two rows). The first and third rows show the neural response (mean firing rate ± standard error on the mean, n = 10 trials) as a function of time. The second and fourth rows show the spike rasters. Within these rows, each panel depicts the spike activity over each of the 10 trials (rows) and time (x-axis), and are color-coded by tested body site. The vertical line labeled 'Go' indicates the start of the stimulus phase. (C) Confusion matrix of the cross-validated classification accuracy (as percentage) for predicting body parts from population neural data. Colors represent the cross-validated accuracy, as in the scale. The matrix is an average of the confusion matrices computed for each recording day individually (*Figure 1—figure supplement 3*). L: left body, ipsilateral to implant; R: right body, contralateral to implant.

The online version of this article includes the following figure supplement(s) for figure 1:

**Figure supplement 1.** Nonstationary waveforms across days indicate that recorded neurons are partially distinct.

**Figure supplement 2.** Postcentral-intraparietal (PC-IP) population responsiveness is not qualitatively changed by pooling together single and multi-units.

**Figure supplement 3.** Population classification statistics were qualitatively unchanged between data recording sessions.

the piecewise linear fit (dashed line). A bootstrap procedure was used to find the interquartile range of latency estimates (*Figure 3B*).

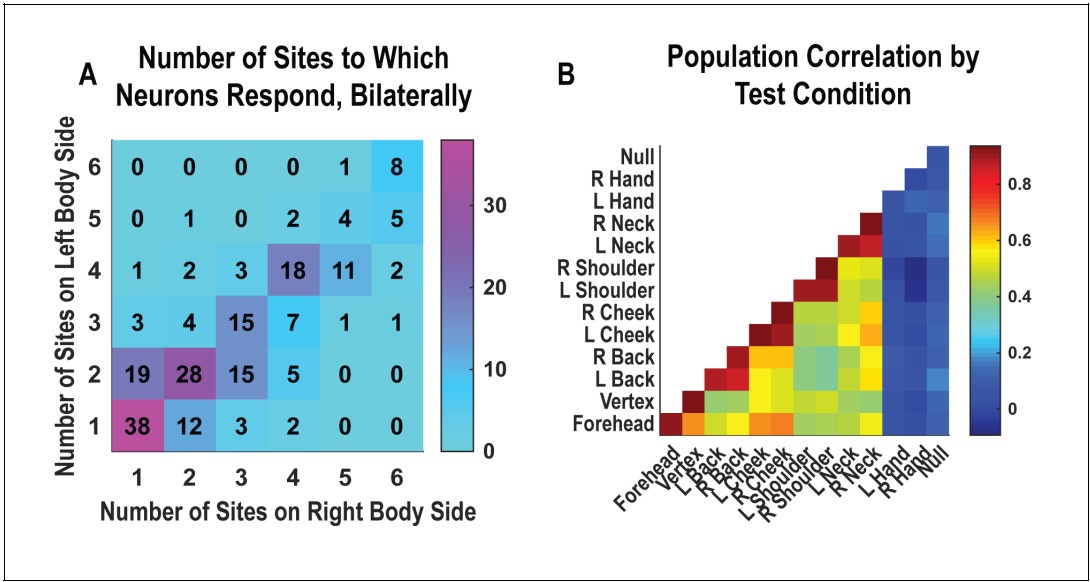

**Figure 2.** Neurons respond to variable numbers of bilateral receptive fields. Bilateral responses are mirror-symmetric. (A) Matrix showing the number of neurons from within the postcentral-intraparietal population that responded to the number of body parts shown, along each of the left and right body sides. Colors represent the number of neurons. Population results were not qualitatively affected by pooling together single and potential multi-units (*Figure 2—figure supplement 1*). Analysis of tactile receptive fields is shown in *Figure 2—figure supplement 2* and *Figure 2—figure supplement 3*. (B) Neuronal population correlation demonstrating the relation in encoding structure between body locations. Colors represent strength of correlation, as in the scale. Population results were not qualitatively affected by pooling together single and potential multi-units (*Figure 2—figure supplement 4*) or by choice of distance metric (*Figure 2—figure supplement 5*). For mirror-symmetry analysis at the single unit level, see *Figure 2—figure supplement 6*. L: left body, ipsilateral to implant; R: right body, contralateral to implant.

The online version of this article includes the following figure supplement(s) for figure 2:

**Figure supplement 1.** Bilateral responses to actual touch are not qualitatively changed by pooling together single and multi-units.

**Figure supplement 2.** Diverse tactile receptive fields in human postcentral-intraparietal (PC-IP) neurons.

**Figure supplement 3.** Receptive fields demonstrate local spatial structure.

**Figure supplement 4.** Symmetry in population-level responses to bilateral touch is not qualitatively changed by pooling together single and multi-units.

**Figure supplement 5.** Relationships of population responses to touch are conserved across distance metrics.

**Figure supplement 6.** The majority of postcentral-intraparietal neurons code tactile receptive fields bilaterally in a mirror-symmetric manner.

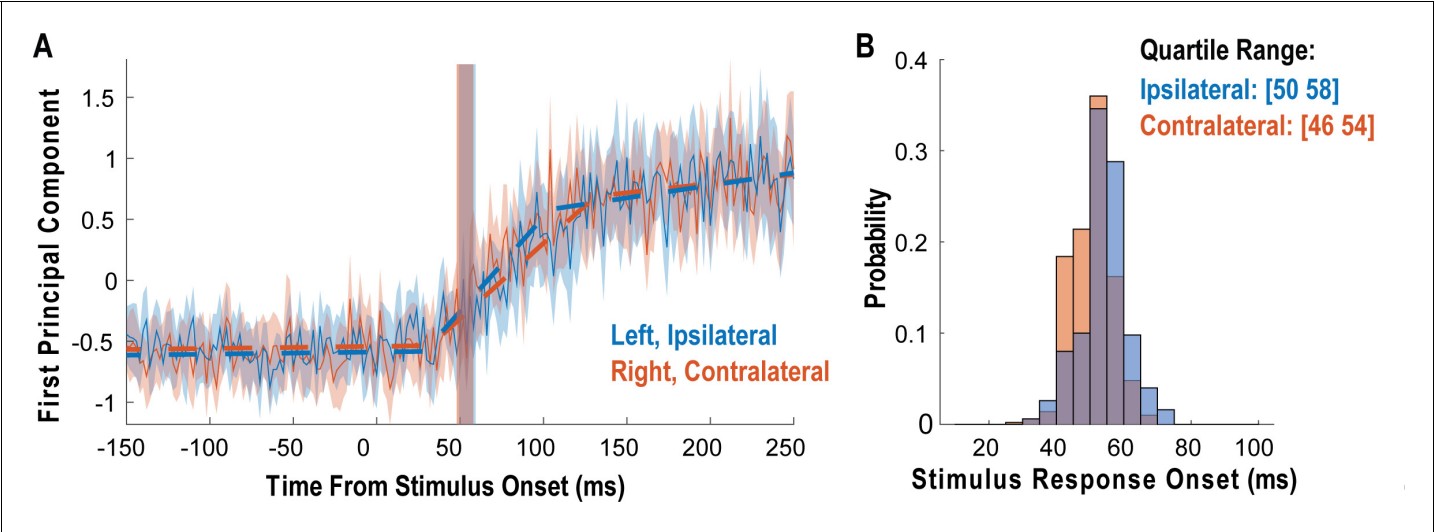

**Figure 3.** Tactile responses occur at short latency. (A) Population response was quantified as the first principal component (mean ± 95% CI). Population response was computed separately for the left (blue; ipsilateral to implant) and right body sides (orange; contralateral to implant) and is shown as a function of time (2 ms window size, 2 ms step size, no smoothing). Dashed lines show piecewise linear fit used to compute latency. Transparent vertical bar shows inter-quartile latency range based on a bootstrap procedure (see B). (B) Distribution illustrating variability of latency estimates for the recorded data using a bootstrap procedure. Color code as in A.

## Tactile imagery task evokes body part-specific responses congruent with actual touch

The results thus far establish that PC-IP neurons have spatially structured tactile receptive fields that are activated at short latency consistent with processing of tactile sensations. Are neurons that encode tactile sensations also recruited during tactile imagery? And if so, how might evoked neural responses compare to those arising from actual touch? To address these questions, we performed an additional experiment allowing us to compare population activity elicited during a tactile imagery task with activity elicited during actual touch to matching body parts recorded during interleaved trials. During the imagery conditions, the participant was instructed to imagine touch to the right (contralateral) cheek, shoulder, or hand with the same qualities as the actual touch stimuli the participant experienced during interleaved trials. A null condition was included as a baseline to measure neural activity when no stimulus was presented.

As with findings for actual touch, neuronal responses elicited during the tactile imagery task following the go cue (during the imagery phase) were discriminably encoded (*Figure 4A*, cross-validated accuracy 92%). High decode accuracy is consistent with the participant's compliance with task instructions and implies that the tactile imagery task elicited discriminative neural responses. A significant fraction of PC-IP neurons encoded actual touch to the cheeks and shoulders but not to the hands (*Figure 4B*; $\chi^2(1)=355.73$, p<0.05), consistent with the results presented in *Figure 1*. In comparison, a smaller fraction of the neuronal population was responsive to the cheek and shoulder during imagery of tactile stimuli (*Figure 4B*). Of note, a significant number of neurons were active during imagined touch to the hand ($\chi^2(1)=188.89$, p<0.05), despite the hand being clinically insensate in the study participant (and despite actual touch to the hand not eliciting neuronal activation). The extent of overlap between the set of units active during actual and the tactile imagery condition is illustrated in *Figure 4C*. The degree of overlap, compared to what is expected by chance, was statistically significant (permutation shuffle test, p<0.05). Results were qualitatively similar for well-isolated single units (*Figure 4—figure supplement 1*).

We used the population correlation measure to compare population-level neural activity across conditions (*Figure 4D*). Neural activity during the tactile imagery task shared a neural substrate with responses evoked by actual touch: representations evoked during the imagery task and during actual touch were more similar for matching body parts than for mismatched body parts (*Figure 4E*, permutation shuffle test, p<0.05).

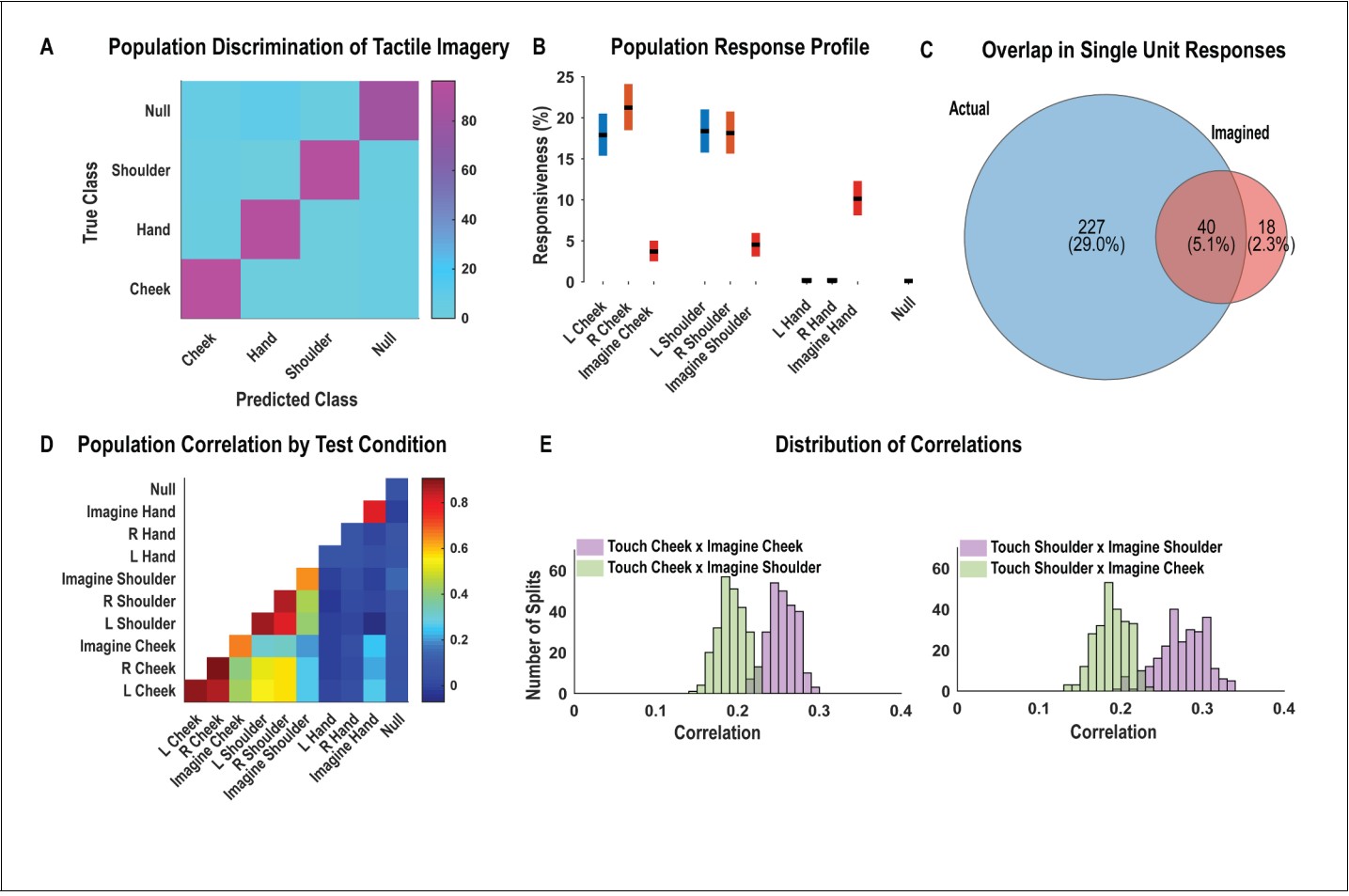

**Figure 4.** Postcentral-intraparietal (PC-IP) neurons encode body part-specific responses during the tactile imagery task. (A) Average classification confusion matrix across recording sessions for body parts during tactile imagery and the baseline (null) condition. Colors represent prediction accuracy as a percentage, as in the scale. (B) Percentage of PC-IP neurons significantly modulated from baseline (mean ± 95% CI, p<0.05, false discovery rate corrected, n = 838 units) split by test condition. Population results were not qualitatively affected by pooling together single and potential multi-units (*Figure 4—figure supplement 1*). (C) Venn diagram illustrating the number (percentage) of PC-IP neurons recorded that activated during actual and imagined touch, and their overlap. (D) Population correlation matrix depicting similarity of the population response between all test conditions. Colors represent the correlation strength, as in the scale. (E) Distribution of correlations between actual shoulder (left) and cheek (right) touch and imagined cheek/shoulder touches, with the distributions computed over different splits of the data (see 'Materials and methods: Population correlation'). L: left body, ipsilateral to implant; R: right body, contralateral to implant.

The online version of this article includes the following figure supplement(s) for figure 4:

**Figure supplement 1.** Postcentral-intraparietal (PC-IP) responses during the tactile imagery task are not qualitatively changed by pooling together single and multi-units.

## Dynamic evolution of population coding between task epochs suggests multiple cognitive processes

The analyses above were restricted to the mean neuronal activity following the go cue (e.g., during actual touch or during imagery) to allow a direct comparison with the results reported for the previous paradigms. We now expand this analysis. During the tactile imagery task, the participant heard a verbal cue specifying a body part (verbal cue = 'cheek,' 'hand,' or 'shoulder') followed approximately 1.5 s later by a beep instructing the participant to imagine the stimulus at the cued body part on the right side of the body. This cue-delay paradigm is standard in the motor physiology literature and is used to dissociate planning from motor execution-related neural activity (*Aflalo et al., 2015*; *Rosenbaum, 1983a*; *Lecas et al., 1986*; *Ames et al., 2019*). In our case, the cue-delay was unique

to the tactile imagery condition. We utilized the cue-delay task to begin to dissociate in time neural activity related to different aspects of the task.

To leverage the benefits of the cue-delay paradigm, we performed a dynamic classification analysis (500 ms windows, stepped at 100 ms). Results are shown as a matrix (*Figure 5*). In brief, the diagonal elements represent the cross-validated prediction accuracy for a specific time window. The off-diagonal elements represent how well the classifier generalizes to alternate time windows. Each row can be interpreted as quantifying how well decision boundaries established for the diagonal time windows generalize to other time windows. This analysis allows us to measure when the neuronal population represents the different body parts (the diagonal) and whether population coding is similar or distinct during the task phases (the off-diagonal). We are interested in two main phases of the task: the early portion comprised the cue and delay (cue-delay), and the later portion when the participant is actively imagining the stimulus (go/imagery). *Figure 5A* schematically illustrates the examples of possible results. The examples are meant to be illustrative and are not an exhaustive list of possibilities. The population may be discriminative exclusively during the imagery phase, during the cue-delay and imagery phases but with distinct population coding, during the cue-delay and imagery phases with identical coding, or during the cue-delay and imagery phases with partially shared and partially distinct coding. Each pattern would suggest a different interpretation of various forms of cognitive processing that may be engaged in a tactile imagery task (see 'Discussion').

The results of our classification analysis (*Figure 5B*) are most consistent with body part selectivity during both the cue-delay and imagery phases, with partially shared and partially distinct population coding of the body parts between phases. The shared component is evident in the significant generalization accuracy in the off-diagonal elements, a representative row of which is shown in *Figure 5C* (blue portion) where cross-validated accuracy generalizes from approximately 70% within the cue-delay phase to approximately 60% during the imagery phase. The distinct population activity between phases is highlighted by a cross-validated Mahalanobis distance that provides a sensitive measure of change that is masked by the discretization process of classification (expanded rationale in 'Materials and methods: Temporal dynamics of population activity'). The findings demonstrate a significant change between the activity patterns in the cue-delay and imagery epochs (*Figure 5C*, gray).

To further clarify the properties of individual units, we conducted a dynamic classification analysis for each recorded unit. This resulted in the same matrices described above, but now each matrix represents how information coding evolves for a single unit. Time-resolved classification data were then analyzed using PCA, the first three principal components of which are shown in *Figure 5D*. A majority of variance (26%) is explained by units that are active during both epochs with similar coding. Coding during the imagery epoch exclusively or during the cue-delay epoch exclusively explained an additional 9% of variance.

## Cognitive processing during the cue-delay and imagery epochs of the tactile imagery task shares a neural substrate with that for actual touch

Finally, we look at how encoding patterns through time generalize between the tactile imagery and actual touch conditions. A dynamic correlation analysis was applied both within and across the imagery and actual touch condition types (*Figure 6A*). In brief, the neural activation pattern elicited to each body location was quantified as a vector, and these vectors were concatenated to form a population response matrix for each condition type and for each point in time. These vectors were then pairwise correlated in a cross-validated manner so that the strength of correlation between conditions could be assessed relative to the strength of correlation within condition and across time. We found that the neural population pattern that defined responses to actual touch was similar to population responses both during the cue-delay or the imagery phases of the imagery task (*Figure 6A*). This implies that cognitive processing prior to active imagery as well as during imagery shares a neural substrate with actual touch. Sample neuronal responses that help to understand single unit and population behavior are shown in *Figure 6B*.

## Discussion

We have previously reported that human PPC encodes many action variables in a high-dimensional and *partially mixed* representation (*Aflalo et al., 2020*; *Zhang et al., 2017*; *Zhang et al., 2020*).

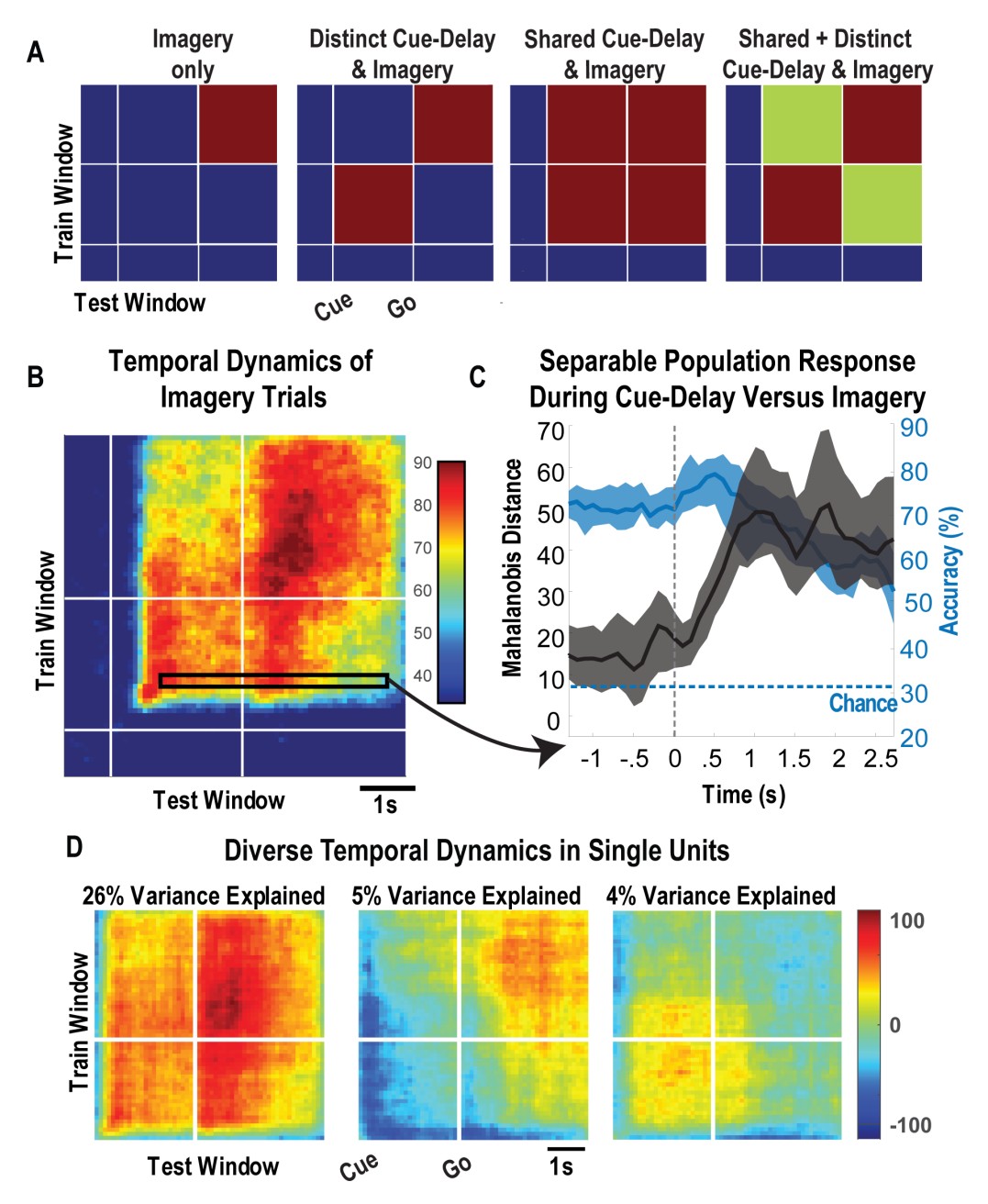

**Figure 5.** Shared and distinct coding of body parts during cue-delay and imagery epochs. (A) Schematic illustrating possible dynamic classification patterns over epochs of the tactile imagery task. In each panel, the window used for classifier training is along the y-axis and the window used for classifier testing is along the x-axis. The start of the auditory cue (marking the onset of the cue-delay epoch) and the beep (marking the go signal for the imagery epoch) is shown as solid white lines, labeled 'Cue' and 'Go.' (B) Dynamic classification analysis results for the imagined touch test conditions with conventions as in A. The colors represent prediction accuracy values (as percentage), as in the scale. (C) Illustration of distinct and shared neuronal responses between the cue/delay and imagery epochs for the boxed window of B. Shared response illustrated with cross-validated, classification generalization accuracy (blue, mean with 95% confidence interval computed across sessions). Distinct response illustrated with cross-validated Mahalanobis distance (gray, mean with 95% confidence interval computed across sessions). The dashed vertical line marks the onset of the imagery epoch. The dashed horizontal line marks chance classification accuracy. (D) Dynamic classification matrices were constructed separately for all selective units. The first three principal components (PCs) of the dynamic classification matrices of single-unit activity are shown, along with the fractional variance explained by each. The mean activity of all neurons within the PC is shown within each panel, color-coded by PC weights. Plot conventions are as in A and B. s: seconds.

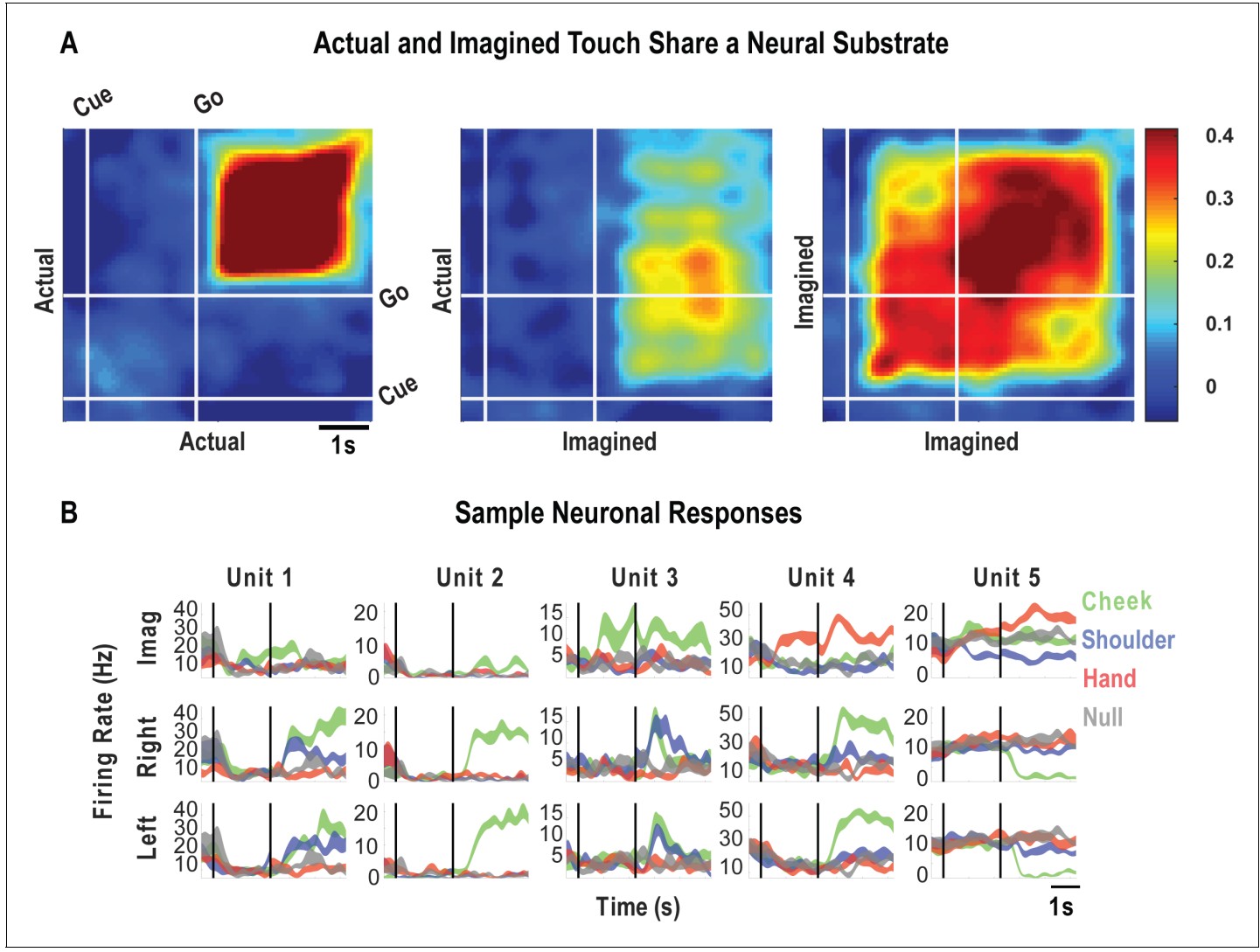

**Figure 6.** Cue-delay and imagery-evoked neural activity shares a neural substrate with actual touch. (A) Within- and across-condition dynamic, cross-validated correlation analysis demonstrating a shared neural substrate between imagined and actual tactile sensations. Each panel shows how the neural population response at one slice of time compares to all other slices of time for the two formats being compared (x- and y-axis labels). Correlation magnitude is indicated by color as in the bar. The start of the auditory cue (marking the onset of the cue-delay epoch) and the beep (marking the go signal for the imagery epoch) is shown as solid white lines, labeled 'Cue' and 'Go.' (B) Representative neuronal responses illustrating selectivity during actual and imagined sensations. Each panel shows the firing rate (in Hertz, mean ± SEM) through time (vertical lines signal onset of cue/delay and go phases as labeled). Each column illustrates the responses of the same unit to tactile imagery of the right side (top), actual touch on the right side (middle), and actual touch on the left side (bottom) for matched body parts.

This architecture allows many parameters to be encoded by a small number of neurons, while still enabling meaningful relationships between variables to be preserved. Here, we show that neurons recorded from the same electrode array in the same clinical trial participant are also selective for bilateral touch at short latency. Responses to actual touch are organized around body part, sharing population representations between the left and right sides. Additionally, a tactile imagery task elicits body part-specific responses that share a neural substrate with that for actual touch. Furthermore, we found neural selectivity during the active imagery epoch as well as during the cue and delay epochs that precede imagery. The distinguishable population activity during these different phases indicates an encoding of multiple cognitive processes that may include semantic association, memory, attention, sensory anticipation, or imagery per se.

## Human PC-IP encodes tactile stimuli with large and bilateral receptive fields

Cortical processing of somatosensory information begins in the anterior portion of the parietal cortex (APC) within four cyto-architectonically defined areas termed BA 3a, 3b, 1, and 2 (*Kaas, 1983*; *Kaas et al., 1979*; *Sur et al., 1980*). Each of these four sub-regions represents primarily contralateral somatosensory information (*Ferezou et al., 2007*; *Geyer et al., 2000*; *Iwamura, 1998*; *Jiang et al., 1997*; *Ruben et al., 2001*; *Schnitzler et al., 1995*; *Tamè et al., 2016*, *Tamè et al., 2015*). Moving from the APC to superior regions of the PPC, spatially localized and segregated sensory representations become progressively more integrated, resulting in neuronal receptive fields that are larger, frequently encompassing multiple segments of the body (*de Lafuente and Romo, 2006*; *Sakata et al., 1973*; *Burton and Sinclair, 1990*; *Dykes, 1983*; *Garraghty et al., 1990*; *Pei et al., 2009*; *Pons et al., 1987*; *Saal et al., 2015*; *Soso and Fetz, 1980*) including bilateral encoding (*Schnitzler et al., 1995*; *Tamè et al., 2016*, *Tamè et al., 2015*; *Zhu et al., 2007*). This process of integration is thought to play an integral role in sensory processing for the guidance of movement (*Graziano et al., 2000*; *Mulliken et al., 2008*). Our results, demonstrating that PPC neurons encode mirror-symmetric spatially structured tactile receptive fields at short latency, are consistent with these prior reports. They further provide the first single neuron evidence supporting a role of human PPC in tactile processing. As hypothesized, when comparing the current results with our prior reports, we find that the same PPC neuronal population engaged by high-level motor cognition also encodes actual tactile sensations, providing a common neural substrate for sensory and motor processing (*Aflalo et al., 2020*, *Aflalo et al., 2015*; *Sakellaridi et al., 2019*).

### Short latency tactile responses

In NHPs, reported latency to touch responses in primary somatosensory cortex (S1) from contralateral touch ranges between 19 and 23 ms (*Reed et al., 2010*; *Vázquez et al., 2012*). PPC response latencies to touch are less clear, but neurons in the lateral intraparietal area within NHP PPC orient to visual stimuli at a mean latency of approximately 45 ms (*Bisley et al., 2004*). A recent human-invasive electrocorticographic study reported mean latencies to visual response of approximately 60 ms in PPC (*Regev et al., 2018*) similar to the mean response latencies to visual stimuli within the occipital cortex (*Bisley et al., 2004*; *Regev et al., 2018*). Our own response latency to actual touch of ~50 ms compares well with these data and is consistent with rapid somatosensory processing within PPC for updating internal estimates of the body (*Graziano et al., 2000*; *Mulliken et al., 2008*).

### Tactile imagery dynamically invokes multiple cognitive processes in human PC-IP that share a neural substrate with actual touch

In motor neurophysiology, neural activity related to planning and execution is dissociated in time by introducing a delay between the cue instructing movement and the movement in response to the cue (*Lecas et al., 1986*; *Rosenbaum, 1983b*). We have previously found that such distinctions between planning and execution are preserved during motor imagery paradigms in tetraplegic individuals (*Aflalo et al., 2015*). Here, a similar paradigm allowed temporal dissociation in cognitive processing during tactile imagery. Single units demonstrated three dominant response profiles (*Figure 5D*): (1) a shared selectivity pattern between the cue-delay and imagery epochs, consistent with cognitive engagement during all phases of the imagery task; (2) selectivity exclusively during the cue-delay epoch but not the imagery epoch; and (3) selectivity exclusively during the imagery epoch but not the cue-delay epoch. In a previous study, we found similarly heterogeneous responses during the cue, delay, and imagery epochs for imagined hand grasp shapes (*Klaes et al., 2015*). These single-unit temporal selectivity profiles provide a basis for the population-level findings of generalization in classification results between the cue-delay and imagery epochs (*Figure 5B,C*) but also a separation in neural state-space between these epochs (*Figure 5C*).

The tactile imagery task evoked body part-specific cognitive activity that shared a neural substrate with actual touch within the PC-IP. Activity during imagined touch to the cheek, for example, was more similar in representation to actual touch to the cheek than to actual touch to the shoulder, and vice versa. Interestingly, the overlapping neural representations between actual touch and those elicited during imagery were not limited to the stimulus phase (actual touch and imagery) itself, but also extended to the cue-delay phase of the imagery task. This overlap echoes our recent findings

for shared neural representations between imagined and attempted actions, as well as for shared neural representations between observed actions and action verbs (*Aflalo et al., 2020*; *Zhang et al., 2017*). These studies are consistent with views in which cognition recruits sensorimotor cortical regions (*Binder and Desai, 2011*; *Meyer and Damasio, 2009*; *Miyashita, 2019*; *Patterson et al., 2007*; *Ralph et al., 2017*). We acknowledge that, as with all passive neural recording studies, ours cannot establish a causal role for these neurons in tactile cognition. Understanding the unique contribution of PC-IP neurons within the larger network of brain regions engaged in cognitive touch processing remains to be explored. Nonetheless, our current results provide the first human single-unit evidence of a shared neural substrate between tactile imagery and actual touch.

One concern with the use of all imagery experiments is that participant compliance cannot be externally validated. This raises the possibility that the participant is not performing the task or is performing the task in an unexpected manner. We think this is unlikely for three reasons. First, the subject by the time of this study was well versed in performing cue-delayed paradigms in the motor domain using both motor imagery and overt movements. In *Zhang et al., 2017*, the participant's performance of overt movements was perfect: the participant performed both the correct cued action and the action at the go cue (i.e., no movements began prior to the go cue as validated by measurements of electromyogram activity; *Zhang et al., 2017*). Second, our current pattern of results that includes stable and accurate body part-specific encoding within the cue-delay and imagery epochs, with a shift between epochs, is consistent with the participant performing the task as instructed. At a minimum, it is consistent with the participant's performing two distinct cognitive operations during the two primary phases of the task with remarkable trial-to-trial consistency. Third, evidence for a shared neural substrate between the actual touch and imagined touch conditions indicates that selective responses during the imagery task are related to tactile cognition.

## What does neural processing within human PC-IP during tactile imagery represent?

While our task identifies dynamic engagement of multiple cognitive processes during tactile imagery, it is inadequate to precisely define the cognitive correlates of the observed neural activity. A number of cognitive processes may be engaged during the tactile imagery task including preparation for and/or execution of imagery, engagement of an internal model of the body, semantic processing of the auditory cue, allocation of attention to the cued body location or nature of the upcoming stimulus, and/or sensory memory for the corresponding actual sensation applied by the experimenter.

The precise neural correlates of tactile imagery are unknown, but evidence suggests that both imagined and actual touch may engage the same internal mental representations, or internal models of the body (*Kilteni et al., 2018*; *Schmidt and Blankenburg, 2019*). Support for such a shared representation comes largely from the parallel domain of motor imagery (*Kilteni et al., 2018*). Imagined and actual movements show similarity at the behavioral (e.g., similar duration), physiological (e.g., alteration of heart rate), and neural levels (e.g., activating the same neural substrates) (*Decety et al., 1993*, *Decety et al., 1991*, *Decety et al., 1989*; *Decety and Michel, 1989*; *Lotze and Halsband, 2006*; *Papaxanthis et al., 2002a*, *Papaxanthis et al., 2002b*). These studies have been interpreted as evidence that imagined movements are the simulation of the internal models that track the state of our bodies during movement (*Jeannerod and Decety, 1995*). In powerful support of such a notion, we have shown that populations of neurons in human PPC code motor imagery and overt actions in highly similar ways (*Zhang et al., 2017*). The domain of tactile imagery has been less studied in comparison. However, relevant to the current paper, behavioral evidence has demonstrated that internal models of motor actions can influence sensory perception of touch (*Kilteni et al., 2018*). Further, human neuroimaging studies suggest that overlapping brain regions are activated during both imagined and actual touch, including the PPC (*Schmidt and Blankenburg, 2019*; *Lucas et al., 2015*; *Wise et al., 2016*). This points to not only a shared substrate for the representation of imagined and actual touch, but also to their likely engagement of similar internal models. Because an internal model may be involved in anticipatory or planning activity (and/or related to imagery), it could at least partly explain the pre-stimulus (post-cue, pre-imagination) neural activity we observed.

Another possibility is that the neural activity following the auditory cue in our study represents semantic processing of the cued word. Evidence suggests that a network of brain regions is

activated in processing word meaning, including those involved in processing their higher-order sensory aspects, or motor intentions such as PPC (*Binder and Desai, 2011*; *Meyer and Damasio, 2009*; *Ralph et al., 2017*; *Huth et al., 2016*; *Martin, 2016*; *Pulvermüller, 2013*). Within this framework, semantic processing of the auditory cue (e.g., instructing imagined touch to the cheek) may engage the same population of neurons responsible for the higher-level processing of touch, consistent with our data. In support, we recently reported that action verbs and visually observed actions share a common neural substrate in the same PPC neural populations reported in the current study (*Aflalo et al., 2020*). Results were consistent with automatic semantic processing as distinct from imagery. The current findings would extend possible semantic processing to the tactile domain and demonstrate neuronal selectivity for auditory cues (in addition to written text used in the previous study).

Hearing an auditory cue can direct the study participant's attention to the cued body part. Attention to a stimulated body part has been shown to enhance sensory processing in human neuroimaging (*Johansen-Berg et al., 2000*; *Hämäläinen et al., 2000*; *Puckett et al., 2017*; *Roland, 1981*). In neurophysiological studies, this manifests as a gain in stimulus responses (*Boynton, 2009*; *Williford and Maunsell, 2006*). However, during the imagery task, no stimulus was delivered to the participant. An attention account of our data would require that attention result in highly discriminable patterns of activity without a sensory stimulus (or pre-stimulus). Most studies of pre-stimulus attention report modest modulation of baseline neural activity (*Boynton, 2009*; *Williford and Maunsell, 2006*; *Snyder et al., 2018*). However, the failure to find pre-stimulus effects may be the consequence of insensitive analysis techniques: indeed, recent single neuron work in NHP visual cortical area 4 (V4) demonstrated discriminable coding for the locus of attention prior to stimulus presentation and, further, that the pre-stimulus activation patterns were systematically related to the post-stimulus response patterns (*Snyder et al., 2018*). These recent results suggest that attention may be decodable elsewhere, and they match the results presented in this paper. It is also consistent with what we have previously described as *partially mixed* selectivity (*Zhang et al., 2017*; *Zhang et al., 2020*). If our results are interpreted within the framework of attention, our current findings are inconsistent with a simple gain-like mechanism for attention, but instead suggest a richer mechanism by which information is selectively enhanced for further processing (*Snyder et al., 2018*).

Our task was not designed to tease apart the different possible cognitive correlates of the observed neural activity engaged during imagery. We think the temporal dynamics of the signal indicate that multiple cognitive process may be engaged throughout the course of the task. The above cognitive phenomena may each independently engage the same neural population as distinct phenomena or may be distinct processes that nonetheless engage the same underlying neural substrate.

## PC-IP and plasticity following spinal injury

The extent to which the human PPC reorganizes following SCI is unknown. Lesion studies in NHPs suggest that BA 3b and 3a, 1 and 2, show altered sensory maps following SCI in a manner dependent on thalamic input from the afferent sensory pathways such as the dorsal column-medial lemniscus system (*Tandon et al., 2009*). With mid-cervical lesions, for instance, there is an initial loss of BA 3b hand representations and a slight expansion in face representation at approximately 2 years (*Tandon et al., 2009*; *Mohammed and Hollis, 2018*). Although significant axonal sprouting has been demonstrated to occur at the site of deafferentation in the spinal cord, with increased projections to brainstem nuclei, the changes observed in the somatosensory cortex are significantly smaller (*Tandon et al., 2009*; *Mohammed and Hollis, 2018*). Moreover, the reorganization in higher-order somatosensory centers such as the secondary somatosensory cortex is even more restricted than in BA 3b (*Mohammed and Hollis, 2018*). Similar stability in the topography of the somatosensory cortex has been identified in human subjects that have suffered limb amputations. In these amputees, there is a preserved digit map within the primary somatosensory cortex (*Kikkert et al., 2016*; *Makin and Flor, 2020*).

The results of our experiments suggest significant stability in tactile somatosensory architecture within the PPC. A substantial fraction of the neuronal population responded to imagined touch to the hand, while there was no significant response to actual touch (insensate in the study participant). The response during the imagery task lends support to the idea that despite the lack of peripheral input from the hand due to the participant's SCI, the brain maintains an internal

representation of tactile sensations (*Makin and Bensmaia, 2017*). The findings that intracortical microstimulation produces discernible tactile perceptions from insensate body regions further reinforce a maintained representation of somatosensation after deafferentation (*Armenta Salas et al., 2018*; *Flesher et al., 2016*). These findings will prove useful for bidirectional neural prostheses. We acknowledge that while additional work probing cortical reorganization following SCI is necessary to fully understand its electrophysiological consequences, our results provide insight into the maintenance of basic tactile processing within the human PC-IP, after SCI.

## Conclusion

Multiple lines of evidence indicate a critical role for the human PPC in the integration of convergent multimodal sensory information to enable complex cognitive processing and motor control. To date, however, its processing of somatosensory information at the single neuron level has remained fundamentally unexplored. In the unique opportunity of a BMI clinical trial, we examined the neural encoding of actual and cognitive touch within the human PC-IP. We found that local populations of PC-IP neurons within a 4 × 4 mm patch of cortex encode bilateral touch sensations to all tested body regions above the level of the participant's injury at short latency. A significant fraction of PC-IP neurons responded during the imagined touch condition with matching sensory fields to actual touch. The activity in the delay period of the task, between cueing and imagining touch, may reflect cognitive processes including tactile semantics, sensory anticipation, attention, as well as active imagery. Together, our results provide the first single-unit evidence of touch processing within the human PC-IP and identify a putative substrate for the encoding of cognitive representations of touch, thus far untested in animal models.

# Materials and methods

**Key resources table**

| Reagent type (species) or resource | Designation | Source or reference | Identifiers | Additional information |
|---|---|---|---|---|
| Software, algorithm | MATLAB | MathWorks, MATLAB R2019b | RRID: SCR_001622 | |
| Software, algorithm | Psychophysics toolbox | Psychophysics toolbox PTB3, http://psychtoolbox.org/ | RRID: SCR_001622 | |
| Other | Neuroport | Neuroport Recording System with Utah array implant, https://blackrockmicro.com/ | | |
| Other | Capacitive Touch Sensory | Adafruit Capacitive Touch HAT for Raspberry Pi, https://www.adafruit.com/product/2340 | Product ID: 2340 | |

## Study participant

The study participant, NS, is a 60-year-old tetraplegic female with a motor complete SCI at cervical level C3-4 that she sustained approximately 10 years prior to this report. She has intact motor and sensory function to the level of her bilateral deltoids. NS was implanted with two 96-channel Neuroport Utah electrode arrays (Blackrock Microsystems model numbers 4382 and 4383) 6 years post-injury for an ongoing BMI clinical study. Implants were made in the left hemisphere as human neuroimaging studies have typically reported stronger coding of intention-related activity in left versus right PPC (*Gallivan et al., 2011a*, *Gallivan et al., 2011b*). She consented to the surgical procedure as well as to the subsequent clinical studies after understanding their nature, objectives, and potential risks. All procedures were approved by the California Institute of Technology (IRB #18–0401), University of California, Los Angeles (IRB #13–000576-AM-00027), and Casa Colina Hospital and Centers for Healthcare (IRB #00002372) Institutional Review Boards.

## Implant methodology and physiological recordings

The electrode implant methodology in NS has been previously published (*Aflalo et al., 2015*; *Zhang et al., 2017*; *Zhang et al., 2020*). One array was implanted at the junction of the left IPS with the left post-central sulcus in what we refer to as PC-IP. The other was implanted in the left superior parietal lobule (SPL). Implant locations were determined based on preoperative fMRI. The

participant performed imagined hand reaching and grasping movements during a functional MRI scan to localize limb and hand areas within this region. Following localization, a craniotomy was performed on August 26, 2014. The PC-IP electrode array was implanted over the hand/limb region of the PPC within the dominant (left) hemisphere, at Talairach coordinates [−36 lateral, 48 posterior, 53 superior]. In the weeks following implantation, it was found that the SPL implant did not function. Although this electrode array was not explanted, only data recorded from the PC-IP implant were used in this study.

## Experimental setup

All experimentation procedures were conducted at Casa Colina Hospital and Centers for Healthcare. Participant NS was seated in a motorized wheelchair in a well-lit room. Task procedures are presented in detail in the sections below. For most tasks, however, one experimenter stood directly behind the participant and was responsible for providing tactile stimuli to the participant. A 27-inch LCD monitor was positioned behind NS (visible to the experimenter but not to the subject) to cue the experimenter for the presentation of a stimulus. Cue presentation was controlled by the psychophysics toolbox (*Brainard, 1997*) for MATLAB (MathWorks) (*Brainard, 1997*).

## Data collection and unit selection

Data were collected over a period of approximately 8 months in the fourth year after NS was implanted. Study sessions were conducted between two and three times per week, lasting approximately 1 hr each. Neural activity recorded from the array was amplified, digitized, and sampled at 30 kHz from the electrodes using a Neuroport neural signal processor (NSP). The system has received Food and Drug Administration (FDA) clearance for less than 30 days of recording. We received an investigational device exemption (IDE) from the FDA (IDE #G120096, G120287) to extend the implant duration for the purposes of the BMI clinical study. Putative neuron action potentials were detected at threshold crossings of −3.5 times the root-mean-square of the high-pass filtered (250 Hz) full bandwidth signal. Each individual waveform was made of 48 samples (1.6 ms) with 10 samples prior to triggering and 38 samples after. Single- and multi-unit activity was sorted using Gaussian mixture modeling on the first three principal components of the detected waveforms. The details of our sorting algorithm have been previously published by our group (*Zhang et al., 2017*). To minimize noise and low-firing effects in our analyses, we used as a selection criterion for units, a mean firing rate greater than 0.5 Hz and a signal-to-noise ratio (SNR) greater than 0.5. We defined SNR for the waveform shapes as the difference between their mean peak amplitude and the baseline amplitude, divided by the variability in the baseline.

Each recording session was assumed to be independent in reporting the total number of units. However, it is likely that some fraction of units recorded on separate days are resamples of the same neuron, given that recordings for different days were made from the same array. Neural waveforms are largely a function of the geometry of the electrode with respect to the neuron, and not a unique signature that can be used to characterize a neuron. Thus, it is impossible to determine whether waveforms collected on separate days represent the same or different neurons. However, the degree to which recordings change from one day to the next can be taken as a general indicator of whether there may be some daily change in which neurons are recorded from. In other words, if the set of waveforms across the array are the exact same from one day to the next, it is likely that we are recording from the same neurons. Conversely, if the waveforms change substantially from one day to the next, it is likely that we are recording from, at least partially, distinct populations of neurons. We performed two analyses to quantify changes in neural recordings across days. In the first, more conservative of the two analyses, we compare the number of waveforms on each channel between days. If the number of waveforms on a single channel changes, then this is strong evidence that there has been some substantial change in the neural recordings. By this measure, an average of 29 ± 4.2% of channels change across days. In the second analysis, we used a permutation shuffle test to measure whether the recorded waveforms on the same channel were more similar than the waveforms across different channels. By this assessment, 58 ± 8% of neurons were distinct from one day to another. These values indicate that there was some degree of neural turnover from one day to the next.

Well-isolated single and multi-units were pooled across recording sessions. To ensure that such pooling did not bias the conclusions of this paper, we performed core analyses within this paper on single units alone, potential multi-units alone, and all units together. The results of these analyses, shown as supplemental figures for key results, generally demonstrate that our results were robust to the pooling of all sorted units together. For the separation of spike sorted units into high-quality single and multi-units, we used as a threshold the mean across all units of the base-10 logarithm of their cluster isolation distances, based on a previously described method (*Zhang et al., 2017*; *Harris et al., 2016*). Sorted units for which the cluster isolation distance was above this measure were considered single units, and those with a distance below this threshold were considered potential multi-units. Findings were robust to the exact choice of isolation distance threshold.

For measurements of neural latency to stimulus response (please refer to the task descriptions below for more information), a custom capacitive probe was used to record the exact time of tactile stimulation. This probe was built using a Raspberry Pi 2B and Adafruit Capacitive Touch Hat (Adafruit product ID 2340). The digital output (a binary output for touch or no touch) was transmitted through a BNC cable into the NSP at an analog signal sampling rate of 2 kHz.

## Task procedures

We used several experimental paradigms to probe various features of actual and imagined touch representations in the PC-IP. In each paradigm, the participant was instructed to keep her eyes closed. The basic task structure comprised three phases. Each trial began with the presentation of a cue to the experimenter (or an auditory cue in the tactile imagery condition, see specific task description below), 1.5 s in duration, indicating the stimulus (e.g., touch NS's left cheek). This was followed by a brief delay, 1 s in duration. Then written text appeared on the screen to signal the experimenter to present the instructed stimulus, for 3 s (in the tactile imagery paradigm, a beep indicated the 'go' signal for the participant). Exact time intervals varied depending on task. Trials were pseudorandomly interleaved; all conditions were necessarily required to be performed at least once before they were repeated. In tasks in which both left and right body sides (ipsilateral and contralateral to the implant, respectively) were tested, stimuli were delivered to one body side at a time.

## Neural responsiveness to touch

This task variant explored neuronal responsiveness and selectivity to actual touch to body parts (receptive fields) with preserved somatosensory input (above the level of SCI). Body parts tested included the forehead, vertex of the head, left and right back of the head, left and right cheeks, left and right sides of the neck, and the dorsal surfaces of the left and right shoulders. As controls, the left and right hands (clinically insensate) and a null condition (no stimulus presentation) were also included. Actual touch stimuli were presented to each body part as finger rubs by the experimenter at approximately one per second. The experimenters' fingertips were used to provide touch stimuli over an approximately 3 × 5 cm patch of skin on each tested body part. Stimuli to the left and right body sides were delivered on separate trials to evaluate each side independently. To ensure that any neural activity observed arose from actual touch and not from observed touch or other stimuli, NS was instructed to close her eyes throughout the task. She additionally wore ear plugs to block auditory input. This task was performed on four separate days. On each day, 10 trials per condition were conducted. In total, we recorded from 398 sorted units on four separate testing days.

## Neural response latency

The purpose of this task was to determine the latency of neural response to actual touch for the left and right sides of the body. Tested regions included the left and right cheeks, the left and right shoulders, and as controls, the left and right hands (insensate). Actual touch stimuli were presented as in the task above. Instead of finger rubs, however, a capacitive touch probe was used to enable precise delineation of the actual time of contact (touch) before the onset of a neural response. This task was performed on eight separate days, with eight trials per condition in each run of the task. In total, we recorded from 838 sorted units.

## Receptive field size

This task aimed to estimate the size of neuronal receptive fields to actual touch. Neural responses to nine equally spaced points were evaluated, 2 cm apart, along a straight line from NS's right cheek to her neck (*Figure 2—figure supplement 3*). Only the right side (contralateral to the implant) was tested in this task. The first of these nine points was on the malar eminence, and the ninth point was on the neck as shown. In addition to the nine points, a null condition (no stimulus presentation) was also included. Stimuli were presented via a paintbrush (3 mm round tip) gently brushed against each location, at a frequency of one brush per second. Touch was restricted to a small region of approximately 0.5 cm (parallel to test sites) by 2.5 cm (perpendicular to the test sites) immediately above the marked points shown in panel A of *Figure 2—figure supplement 3*. The paintbrush was employed to deliver spatially localized sensations without accompanying skin distortion that could mechanically stimulate nearby sensory fields. Data were recorded on six separate days. On each day, 10 trials of each condition were tested. In total, we recorded from 585 sorted units.

## Engagement during tactile imagery

This task was intended to establish whether PC-IP neurons are engaged by tactile imagery and whether neural patterns evoked by cognitive processing of imagined touch and actual touch share a common neural substrate (e.g., activate the same population of neurons in similar ways). In this variant, the participant was presented with either actual touch stimuli or instructed to imagine the sensation of being touched. NS was instructed to keep her eyes closed throughout. Actual stimuli were cued to the experimenter with written words that appeared on the monitor. Because the participant's eyes were closed, the participant did not receive any information about the body part that would be stimulated prior to experiencing the touch. The cue was followed by a 1 s delay, and then at the sound of a beep (the 'go' signal), rubs at 1 Hz were presented with the capacitive touch probe to either the left or right cheeks, shoulders, or hands. During imagined touch trials, an auditory cue was presented to NS instructing her to imagine being touched on her right cheek, shoulder, or hand. The auditory cue consisted of a voice recording of the words 'cheek,' 'hand,' or 'shoulder' with cue duration of approximately 0.5 s. After a 1 s delay, at the sound of the beep, NS imagined touch to the cued body part. We asked the participant to imagine the sensations as alternating 1 Hz rubbing motions similar to what she actual during actual touch trials. A null condition (without actual or imagined touch), not preceded by an auditory cue, was used to establish a baseline neural response. Data were recorded on eight separate days. Eight trials of each condition were performed on each testing day. In total, we recorded from 838 sorted units.

## Quantification and statistical analysis

In the analysis of data from the various task paradigms used in this study, we utilized several statistical methods. Some were specific for certain tasks, but others were applicable to multiple sets of data from the different paradigms. For ease of reference, we have described all methods together in this section. Where necessary, we provide specific examples from tasks to help illustrate their use in this paper. Unless explicitly noted, all recorded units for a given task were used in the statistical analyses pertaining to that task.

### Linear analysis (relevant for *Figures 1* and *4*, and for *Figure 1—figure supplement 2*, *Figure 2—figure supplement 3*, and *Figure 4—figure supplement 1*)

To determine whether a neuron was tuned (i.e., differentially modulated to touch locations), we fit a linear model that describes firing rate as a function of the neuron's response to each touch location. Neuronal responses were summarized as the mean firing rate computed between 0.5 and 2.5 s after stimulus presentation onset. The starting time of 0.5 s was chosen to minimize the influence of variable experimenter delay in presenting the stimulus. The baseline response was summarized as the mean firing rate during the 1.5 s window before the stimulus presentation cue. The linear equation is written as

$$FR = \sum_c \beta_c X_c + \beta_0$$

where $FR$ is the firing rate, $X_c$ is the vector indicator variable for touch location $c$, $\beta_c$ is the estimated scalar weighting coefficient for touch location $c$, and $\beta_0$ is a constant offset term. $X_c$ was constructed by assigning a value of 1 if the corresponding firing rate was collected when touch location c was being stimulated and with a 0 otherwise. All baseline samples were also assigned a 0, effectively pooling together baseline data independent of subsequent touch location. Here, we used indicator variables as our predictors because our stimulus was applied in a binary manner, either touch was applied to a position on the skin or not. Note that in principle the formalism of linear models allows multiple indicator variables to take on a value of 1 at the same time. In our experiment, this would amount to simultaneous touch of two or more body parts. However, in out experiments, simultaneous touch was not tested and thus only one indicator variable could take a value of 1 at a time. Neural responses to a particular body location were considered responsive if the t-statistic for the associated beta coefficient was significant ($p < 0.05$, FDR corrected for multiple comparisons). A unit was considered tuned if the F-statistic comparing the beta coefficients was significant ($p < 0.05$, FDR corrected for multiple comparisons).

The linear models for each task were computed using all test conditions within the task, except when comparing discriminative coding between the left and right body sides. For this analysis, the goal was to determine how informative information encoded for one body side was for the other. Each neuron was fit by two linear models, one for touch locations exclusive to sensate regions of the contralateral side (e.g., contralateral shoulder, neck, back, and cheek) and one for touch locations exclusive to sensate regions of the ipsilateral side (e.g., ipsilateral shoulder, neck, back, and cheek). More details regarding this analysis are given in 'Materials and methods: Tests for mirror symmetric neural coding of body locations: single-unit analysis.'

### Population correlation (relevant for *Figures 2* and *4*, and for *Figure 2— figure supplement 4* and *Figure 4—figure supplement 1*)

We used correlation to compare the population neural representations of various tested conditions (stimulus presentations) against each other in a pairwise fashion. Correlation was chosen over alternative distance metrics (such as Mahalanobis or Euclidean distance) because it provides an intuitive metric of similarity that is robust to gross changes in baseline neural activity across the entire neural population. Alternative distance metrics were tested and gave comparable results (e.g., *Figure 2— figure supplement 5*).

To perform the population correlation analyses, we quantified the neural representations as a vector of firing rates, one vector for each condition (stimulus location) with each vector element summarizing the response of an individual unit. As before, neural activity was summarized as the mean firing rate during the stimulation phase window, defined as 0.5–2.5 s after the onset of stimulus presentation. Firing rate vectors were constructed by averaging the responses across 50–50 splits of trial repetitions. The mean responses across different splits were correlated within and across conditions (e.g., across stimulations of different sensory fields), then the splits were regenerated, and the correlation computed 250 times. Performing the splits 250 times was chosen based on an empirical analysis applied to preliminary data. For preliminary data, we performed the analysis with N splits, with N ranging from 5 to 200 in steps of 5. We found that the mean correlation across splits converged to a stable value by about 80 splits. We then roughly tripled that to ensure that the numerical sampling scheme would capture a stable value of our cross-validated correlation metric. The across-condition correlations measured similarity between population responses for different sensory fields, answering the question – are the tactile sensations similar or dissimilar from the perspective of the recorded neural population? The within-condition correlations assist in our interpretation of the across-format correlations by allowing us to quantify the theoretical maxima of the similarity measure (e.g., if the within-condition correlation is measured at 0.6, then an across condition of 0.6 suggests identical neural representations).

To test whether the difference between any pair of conditions was statistically significant, we used a shuffle permutation test applied to the correlations computed over the 250 random splits. To illustrate, in *Figure 4E* we applied this analysis to test whether the correlation between actual and imagined cheek touch was significantly different from that of actual cheek touch and imagined shoulder touch. The true difference in the correlations was computed as the difference in the mean correlations between actual and imagined cheek touches (over the 250 splits) and the mean of the

correlations between actual cheek touch and imagined shoulder touches. We then randomly shuffled the two distributions together (2000 times) and computed the difference in the mean correlations for each shuffle. The distribution of shuffled differences served as the null distribution, against which we compared the true difference to determine significance. As in the case above, our permutation shuffle test used 2000 shuffles to ensure that the numerical sampling scheme would capture a stable value of the percentile of our true difference compared to the empirical null distribution.

## Decode analysis (confusion matrix; relevant for *Figures 1* and *4*, and for *Figure 1—figure supplement 3*)

Classification was performed using linear discriminant analysis with the following parameter choices: (1) only the mean firing rates differ for unit activity in response to each touch location (covariance of the normal distributions are the same for each condition) and (2) firing rates for each unit are independent (covariance of the normal distribution is diagonal). These choices do not reflect assumptions about the behavior of neurons, but instead were found to improve cross-validation prediction accuracy on preliminary data. In our experiments, we acquired 10 repetitions per touch location, generally not enough data to robustly estimate the covariance matrix that describes the conditional dependence of the neural behavior on the stimulus. In choosing equal covariance, we are able to pool data across touch locations, achieving a more generalizable approximation of the neural response as verified by cross-validation.

The classifier took as input a matrix of firing rates for all sorted units. The analysis was not limited to significantly modulated units to avoid 'peeking' effects (*Smialowski et al., 2010*). Classification performance is reported as prediction accuracy of a stratified leave-one-out cross-validation analysis. The analysis was performed independently for each recording session, and results were then averaged across days.

## Tests for mirror symmetric neural coding of body locations: single-unit analysis (relevant for *Figure 2—figure supplement 6*)

The purpose of this analysis was to assess whether neural responses to one body side were the same as neural responses to the alternate body side on a single-unit basis. Heuristically, we used a cross-validation approach, similar in concept to the population correlation, to ask whether the neural responses to one body are similar to the other body side. The transition to single units required one major modification from the population approach: instead of comparing the pattern of response across neurons (as in the population case), we compared the pattern of response across the set of lateralized body locations (shoulder, neck, cheek, and back). We first selected the set of neurons that demonstrated discriminative encoding to at least one of the body locations that was tested to ensure that there was a meaningful discriminative pattern across sites to form a basis of comparison. Then we used a cross-validation procedure to compare within and across body-side encoding. A schematic representation of how the two sides were compared is shown in panels B–F of *Figure 2—figure supplement 6*.

For each neuron, we created a linear model that explained firing rate as a function of the response to each touch location on the right side. The linear model was constructed using indicator variables as described above; however, the set of body locations was restricted to shoulder, neck, cheek, and back. In this way, the response of a neuron is quantified by the continuous set of beta values for the four locations. This model was then used to predict the responses for the same four locations on the left side. The ability to predict the responses was quantified as the $R^2_{\text{Right to Left}}$. This metric is hard to interpret on its own; a low $R^2_{\text{Right to Left}}$ could indicate that responses are very distinct between the right and left sides or it could indicate that the neuron is not very discriminative (e.g., there is high trial-to-trial variability relative to the differences in response to the different touch locations). Therefore, we also computed a cross-validated $R^2_{\text{Left to Left}}$ measure. This disambiguates the $R^2_{\text{Right to Left}}$ measure. If $R^2_{\text{Right to Left}}$ is low but $R^2_{\text{Left to Left}}$ is high, then we know that the unit is discriminative, but that the patterns of response between the right and left sides are distinct. To compare apples-to-apples, both the $R^2_{\text{Left to Left}}$ and $R^2_{\text{Right to Left}}$ were computed using leave-one-out cross-validation. This is necessary to ensure that the two measures are computed based on the same amount of training data. To directly compare these values, we plotted them against each other

as a scatter plot. If the patterns of response are similar, this would lead to data points falling along the identity line. If the patterns are distinct, the data points should fall below the identity line.

## Response latency (relevant for *Figure 3*)

We quantified the neural response latency to touch stimuli at the level of the neural population. Prior to the analysis, trials were aligned by touch onset as detected by the capacitive touch sensor (ground truth). PCA was used to summarize the population-level temporal response of recorded neurons (*Cunningham and Yu, 2014*). We constructed a matrix of neural data D that was (n) by (t * c * r) in size, with n being the number of neurons, t being the number of time points, c being the number of conditions, and r being the number of trial repetitions. For each neuron, activity was sampled every 2 ms and no additional smoothing was applied. 2 ms windows were chosen to allow high temporal resolution to precisely localize the timing of the neural response with respect to touch contact. We used t = 201 time bins starting from −150 ms and stopping at 250 ms with respect to the time of touch sensor contact. Different ranges from time of contact (up to −500 ms before and 500 ms after probe contact) were tested, and the basic average latency was robust to the exact window choice. c = 2, including data for touch to the cheek and touch to the shoulder. r = 10, as we acquired 10 repetitions per condition. Principal components were calculated based on the singular value decomposition algorithm.

The first principal component (1PC) was retained, and responses were averaged across conditions and repetitions. Single-trial results were visually inspected, and basic temporal profiles were consistent across conditions and repetitions. This process was performed separately for data acquired for touch to the left side and right side of the body. The 1PC was then fit with a piecewise linear function with two transition points. The choice of two transition points was set based on visual inspection of the data and allows for an initial baseline, a subsequent rise, and a final plateau. The time at which transitions occurred was not constrained, being purely a product of the fitting process. Latency was reported as the time the piecewise linear fit crossed the 95th percentile of the baseline data, as measured by the distribution of activity in the window between −150 and 0 ms. To compute bootstrapped quartile bounds of the latency estimates, the above process was repeated 1000 times while resampling with replacement from the 1PC single-trial results. To verify that 1000 resamples were sufficient to estimate a stable estimate of the quartile range, we repeated the process with 1500 resamples and found that the quartile estimate changed less than 1%.

To determine whether the mean difference of latency estimates was significant between the right and the left side, we performed a permutation shuffle test. We used a rank test to compare the true difference in latency estimates against an empirical null distribution of differences in latency estimates generated by shuffling labels and repeating the comparison 2000 times.

## Quantifying macroscale receptive field structure (relevant for *Figure 2—figure supplement 2*)

We found that many neurons responded to touch to multiple body locations. We wished to further characterize the receptive field structure to determine whether neurons were characterized by single-peaked broad receptive fields or discontinuous receptive fields with multiple peaks. To adjudicate between these possibilities, we selected touch locations to the contralateral (right) forehead, cheek, neck, and shoulder for further analysis because these locations are approximately collinear. We reasoned that if neurons are characterized by single-peak-type responses, then responses across a collinear set of testing sites will result in a single local maximum (either with a single peak and fall off on either side or as a monotonic increase to the edge locations). On the other hand, if receptive fields are characterized by multiple peaks, then responses should have multiple local maxima.

Neurons were first restricted to those demonstrating significant differential responses between the four sites (ANOVA, p<0.05, FDR corrected). Each neuron was then grouped according to its location of preferred (peak) response. This resulted in four groups of neurons: neurons that responded maximally to the forehead, cheek, neck, or shoulder. For each neuron, the goal was to identify if the firing rate monotonically decreases with increasing distance from the preferred location or rises again, allowing for a second maxima. For example, for a unit preferring the forehead, this would manifest as firing rate at forehead larger than at cheek, at cheek larger than at neck, and at neck larger than at shoulder. Tests of firing rate between adjacent locations were performed by

one-tailed t-tests between the pair of locations, evaluating whether the firing rate at the location nearer the preferred response was greater than that at the location more distant. In the example of the forehead preferring units, the t-tests evaluated whether cheek>neck and neck>shoulder. If it was found that any of those comparisons was not true (e.g., firing rate at neck greater than at cheek) after correcting for multiple comparisons, this implied a second local maxima. The unit was then classified as multi-peak. If no second local maxima was found, the unit was classified as single peak.

## Receptive field size estimation (relevant for *Figure 2—figure supplement 3*)

In our first experiment, we tested touch responses across major body parts at a course resolution. Patterns of neuronal responses suggest that multiple body parts can be represented in individual neurons, although the response field around each body part is locally narrow (not expansive, covering all body parts). To evaluate this further, as a complimentary dataset, we tested tactile representations at a finer spatial precision to begin to characterize the size of their receptive fields. We characterized the response patterns of individual neurons to tactile stimuli delivered to each of nine points along the subject's face and neck. All units demonstrating a differential spatial response to touch to each of the nine fields were included in this analysis. For each of these units, we first identified the preferred site of stimulus delivery as the point associated with the largest firing rate. Next, we examined its response to delivering stimuli to the other points. To estimate the average size of a neuronal receptive field as a function of its preferred point of stimulus delivery, we fit a Gaussian model to the average responses grouped by the preference of the neuron. The Gaussian model had three free parameters and was defined as

$$G(x) = Ae^{-\frac{1}{2}\left(\frac{x-\mu}{\sigma}\right)^2} + c$$

Here, $A$ is the amplitude of the Gaussian, $\sigma$ is the standard deviation, and $c$ is the constant offset term. $\mu$ is the mean/center of the Gaussian and was fixed at the preferred point. A separate model (with the appropriate value of $\mu$) was fit to each of the response groups. The field size was described as the full width at half maximum.

## Temporal dynamics of population activity during tactile imagery task: within category (relevant for *Figure 5*)

We performed a sliding-window classification analysis to quantify the strength and temporal dynamics of population coding in the tactile imagery task. In this task, the participant heard an auditory cue specifying a body part ('cheek,' 'hand,' or 'shoulder') that lasted approximately 0.5 s, followed by an approximately 2 s delay, and finally a beep instructing the participant to initiate imagining a touch sensation at the cued body part. This task could engage at least four cognitive processes: (1) semantic processing of the cue, (2) preparation/anticipation for imagery, (3) attentional modulation, and (4) imagined touch per se. We used a dynamic classification analysis to understand how the neural population evolved during the course of the trial to determine whether the population was best described as mediating single or multiple cognitive processes. In brief, the analysis consisted of creating a dataset that consisted of the population response measured in small temporal windows throughout the course of the trial. We trained a classifier separately on each temporal window and applied each classifier to both temporal windows. In this way, we can measure how information about the cued stimulus evolves in time (e.g., does there exist neural coding during the delay portion of the trial, and, if so, does the neural coding during the delay match neural coding during active imagery). Classification was performed using linear discriminant analysis as described above. We used cross-validation to ensure that training and predicting on the same time window was directly comparable to training on one window and testing on an alternate time window; in other words, we were careful to ensure that accuracy across all comparisons reflects generalization accuracy using the same amount of training and test data. Classifiers were trained and tested on neural responses to the three imagery conditions: cheek, hand, and shoulder. Population response activity for each time window was computed as the average neural response within a 500 ms window, stepped at 100 ms intervals. Window onsets started at −700 ms relative to auditory cue onset (cue-delay epoch) with the final window chosen 3.5 s after the beep (onset of the imagery epoch).

Classification was performed on all sorted units acquired within a single session. Mean and boot-strapped 95% CIs were computed for each time bin from the cross-validated accuracy values computed across sessions.

We used a fixed window size for averaging time series data for analysis (box-car smoothing) as it provides straightforward bounds for the temporal range of data that are included in the analysis for a particular time window. 500 ms was chosen as a good balance between temporal resolution and noise mitigation. We note that although the window size can influence various metrics (e.g., larger smoothing windows can increase coefficients of determination, $R^2$) the choice of smoothing size is largely inconsequential as long as the kernel size is kept consistent when making comparisons across conditions. The choice of a 100 ms step size was anchored to the choice of smoothing window. A small step, such as 1 ms, would not be justified with a 500 ms time window. We chose 100 ms, representing a change in 20% of the data, to allow us the ability to temporally localize changes in neural response without unnecessarily oversampling a smoothed signal and thus not unnecessarily increasing computation time for analysis.

We believe that this technique, by helping us to understand when information appears and how information compares across task phases, provides a valuable approach to understanding how population activity relates to the underlying cognitive processes. For example, if neural decoding reaches significance only after the go cue, neural activity would be inconsistent with semantic or anticipatory processing. Alternatively, if neural processing begins with the cue, and the same pattern of neural activity is maintained throughout the trial, with no changes during the active imagery phase, then the data would be inconsistent with processing imagined touch per se.

The classification analysis described above was used to measure general similarity of the population response to the tested conditions across time. However, to explicitly test whether population activity was changing, we used Mahalanobis distance as our measure. This is necessary as classification involves a discretization step that makes the technique relatively insensitive to changes in neural population activity that do not cross decision thresholds. Mahalanobis distance, being a proper distance measure, is a more sensitive measure of change. To illustrate, imagine that a classifier is trained on time point A and tested on time point B. At time point A, the means of the two classes are 0 and 1, respectively, and at time point 2 the means are 0 and 4, respectively. All classes are assumed to have equal but negligible variance (e.g., 0.01) in this example. When trained on time point A, the classifier finds a decision boundary at 0.5. with 100% classification accuracy. When tested on time point B, with the same 0.5 decision boundary, the classifier again is 100%. Naively, this could be interpreted as signifying that no change in the underlying data has occurred, even though the mean of the second distribution has shifted.

Separation in neural activity between the cue-delay epoch and the imagery epoch was quantified using a cross-validated Mahalanobis distance computed between the observed neural activity at a time point and a reference (baseline) defined as the neural activity immediately following the presentation of the auditory cue, from 0.25 to 0.75 s. Distances were measured separately for each of the three conditions and then averaged. The mean and standard error on the mean were computed across sessions for the cross-validated distance measures and plotted in *Figure 5C*. Activity during the cue-delay epoch and go epoch was compared using a rank-sum test of the averaged activity during the phase-averaged responses across sessions.

## Temporal dynamics of single-unit activity during tactile imagery task: within category (relevant for *Figure 5*)

We wished to understand the behavior of single neurons that led to the temporal dynamics of the population. The temporal dynamics of single-unit activity during the imagery task (for the imagined touch conditions only) were quantified a PCA (*Figure 5D*). A sliding-window classification analysis was first performed on each sorted unit from all testing days in the same manner as described above for the population activity, with the exception that classifier took as input a vector of the firing rates for a single unit as opposed to a matrix of the firing rates for all units recorded in a single session. This allowed a quantitative description of the temporal dynamics for each sorted unit. Next, a PCA was applied to the dynamic classification matrices with individual neurons counting as the independent observations. PCA has become a standard method for describing the behavior of neural populations (*Cunningham and Yu, 2014*). Typically, PCA is applied to firing rate measurements of

neurons. However, in our case, we were less interested in capturing the main modes of variability with respect to individual conditions, but instead wanted to capture the main modes of variability with respect to the temporal dynamics of information encoding.

### Temporal dynamics of population activity during tactile imagery task: across category (relevant for *Figure 6*)

We wished to evaluate the similarity in neural representations of actual and imagined touch in a time-resolved manner, as well as to compare the similarities in activity from one epoch (cue-delay) to another (stimulus: actual or imagined touch). We performed a sliding-window (dynamic) correlation analysis in a cross-validated manner to compute within-format correlation in addition to across-format correlations. For this analysis, we restricted the tested body sites in the actual touch format to the right cheek and right shoulder only. Neural activity from the left side was not used, to try and match the conditions for the imagined touch format, in which only touch to the right side was tested. Similarly, the hand was not included in this analysis to match conditions that evoked responses in both formats.

For cross-validation, the analysis began with splitting trial repetitions into training and testing sets (five trial repetitions each). A sliding time-window was used for the analysis with window size of 500 ms and step size of 100 ms. Correlations were computed between training and testing sets for all combinations of windows, starting from 500 ms before the cue onset to 1000 ms after the end of the stimulus phase. Within each window, we organized the neural response data into two matrices (one each for the training and the test trial splits) with two columns each. Each column contained trial-averaged firing rates during the corresponding time window for each of the two tested stimulation sites (cheek and shoulder), with one value per unit. The columns represented the two formats. Thus, for N units recorded, two tested stimulation sites and two formats (actual and imagined), each matrix was of size (2*N) $\times$ 2. The mean response across each matrix (computed separately for training and test sets) was subtracted from each value to ensure that a positive correlation across formats reflected a similarity in the pattern of responses to the two body sites and not general offsets in the mean response of the different neurons. Finally, correlations were computed between training and test sets for all combinations of time windows. This was done across 50 random 50–50 trial splits and the results averaged across these repetitions. The analysis was performed for each recording session independently and the depicted results averaged over days.

## Acknowledgements

The authors thank subject NS for participating in the studies, Viktor Shcherbatyuk for technical assistance, and Kelsie Pejsa for administrative and regulatory assistance. This work was supported by the National Institute of Health (R01EY015545), the Tianqiao and Chrissy Chen Brain-machine Interface Center at Caltech, the Conte Center for Social Decision Making at Caltech (P50MH094258), and the Boswell Foundation.

## Additional information

### Funding

| Funder | Grant reference number | Author |
|---|---|---|
| National Eye Institute | R01EY015545 | Tyson Aflalo<br>Nader Pouratian<br>Richard A Andersen |
| Conte Center | P50MH094258 | Tyson Aflalo<br>Richard A Andersen |
| T&C Chen Brain-Machine Interface Center at Caltech | | Tyson Aflalo<br>Richard A Andersen |
| Boswell Foundation | | Richard A Andersen |

The funders had no role in study design, data collection and interpretation, or the decision to submit the work for publication.

## Author contributions
Srinivas Chivukula, Data curation, Software, Formal analysis, Validation, Investigation, Methodology, Writing - original draft, Writing - review and editing; Carey Y Zhang, Data curation, Software, Formal analysis, Investigation, Methodology, Writing - original draft, Writing - review and editing; Tyson Aflalo, Conceptualization, Resources, Data curation, Software, Formal analysis, Supervision, Funding acquisition, Validation, Investigation, Methodology, Writing - original draft, Project administration, Writing - review and editing; Matiar Jafari, Resources, Data curation, Validation, Writing - review and editing; Kelsie Pejsa, Resources, Data curation, Supervision, Funding acquisition, Methodology; Nader Pouratian, Resources, Data curation, Funding acquisition, Methodology, Writing - review and editing; Richard A Andersen, Resources, Supervision, Funding acquisition, Validation, Project administration, Writing - review and editing

## Author ORCIDs
Srinivas Chivukula https://orcid.org/0000-0002-3570-162X
Tyson Aflalo https://orcid.org/0000-0002-0101-2455
Nader Pouratian http://orcid.org/0000-0002-0426-3241
Richard A Andersen https://orcid.org/0000-0002-7947-0472

## Ethics
Clinical trial registration NCT01958086.
Human subjects: All procedures were approved by the California Institute of Technology (IRB #18-0401), University of California, Los Angeles (IRB #13-000576-AM-00027), and Casa Colina Hospital and Centers for Healthcare (IRB #00002372) Institutional Review Boards. Informed consent was obtained after the nature of the study and possible risks were explained.

## Decision letter and Author response
Decision letter https://doi.org/10.7554/eLife.61646.sa1
Author response https://doi.org/10.7554/eLife.61646.sa2

## Additional files
### Supplementary files
• Transparent reporting form

### Data availability
Data and analysis for key figures will be made available on github: https://github.com/tysonnsa/eLifePPCTouch copy archived at https://archive.softwareheritage.org/swh:1:rev:aead504c828568a46cf9555598211f1800f2187d/.

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
