## [Decision Letter]

Thank you for submitting your article "Neural encoding of felt and imagined touch within human posterior parietal cortex" for consideration by *eLife*. Your article has been reviewed by three peer reviewers, and the evaluation has been overseen by Tamar Makin as the Senior and Reviewing Editor. The reviewers have opted to remain anonymous.

The reviewers have discussed the reviews with one another and the Reviewing Editor has drafted this decision to help you prepare a revised submission.

Chivukula and colleagues report an extensive set of multi-unit neural recordings from PPC of a tetraplegic patient taking part in a brain machine interface clinical trial. The recordings were collected across a set of tasks, designed to investigate neuronal responses to both experienced and imagined touch. It was found that many neurons are responsive to touch in specific locations. Most of the recorded neurons were activated bilaterally, which is consistent with earlier monkey work from this lab. Probably the most important component of the work is the analysis of the modest activation in this area that occurs simply when the participant imagines different places on her body being touched – even the insensate arm. This work is virtually impossible to do in animals, and as such offers a unique opportunity to describe neural properties for higher-level representation of touch. The study therefore paves the way for a deeper understanding of the role of the human PPC in the cognitive processing of somatosensation.

Overall, we found the manuscript to be well-written, the study to be interesting, and for the most part the analyses are well thought out. But at the same time, the reviewers raised multiple main concerns regarding missing information and unclear descriptions of some of the analyses undertaken, which are detailed below. In addition, it was felt that there was unnecessary overlap across analyses – the first part especially contains a number of analyses that seem to make very similar points repeatedly or where it is not entirely clear what the point is in the first place. As such, there is a need to identify and cut a lot of the duplicative analyses/results and explain both the essential methods and the interpretation of the remaining results more succinctly and clearly. The key analyses could then be streamlined and better justified, ideally with an eye towards a consistent approach in both parts of the paper. As you will see, most of the comments detailed below are geared towards guiding the authors through this major revision (please forgive any duplicative comments on our behalf, given the technical nature of most of the comments, the Editor was keen to preserve the original comments as made by the reviewers). In addition, there are also some major considerations regarding the contextualisation and interpretation of the key imagery results, as detailed in the first major comment below.

1) Perhaps the most exciting innovation of the study relates to the neural responses related to the imagery conditions. Yet, we know from many previous studies in humans that cognitive processes (most notably attention), can modulate somatosensory responses. What the current study offers is an opportunity to characterise this cognitive modulation at a single neuron (and population) level over space and time, with hopes of achieving better insight into its underlying mechanism. While the reviewers agree that this unique contribution is valuable, it was also agreed that the manuscript would benefit from additional context in the Introduction as well as a more thorough discussion – particularly with respect to related research on this topic and potential mechanisms that could be recruited to support the observed neural modulation during the imagery condition.

Introduction: The second paragraph did well in establishing why one might be interested in examining somatosensory processing in the PPC. It was however, less clear why the particular questions at the end of the paragraph were being posed. Perhaps an extra paragraph could be added to bridge the notion that a sizeable body of literature has been developed around somatosensory representation within the PPC and the several "fundamental" questions remaining that are of interest here.

Discussion: The manuscript would benefit from a more thorough discussion of "imagination per se" and the various top-down processes that might be involved – as well as better positioning with respect to previous studies investigating top-down modulation of the somatosensory system. The authors state that the cognitive engagement during the tactile imagery may reflect semantic processing, sensory anticipation, and imagined touch per se – which we would not argue. But we would also expect some explicit mention of processes like attention and prediction. Perhaps these are intended to be captured by "sensory anticipation" – but, for example, attention can be deployed even if no sensation is anticipated. Importantly, it seems that imagining a sensation at a particular body site might well involve attending to that body part. That is, one may first attend to a body part before "imagining" a sensation there – and then even continue to attend there the entire time the imagining is being done. Because of this, perhaps the authors are considering attention to be a part of "imagination per se". But since attention has been shown to modulate somatosensory cortex without imagination, how can one exclude the possibility that the neuronal activity measured here simply reflects this attention component?

Regardless, we think the Discussion would benefit from a more explicit treatment of these top-down processes – especially given the number of previous studies showing that they are able to modulate activity throughout the somatosensory system. Some literature that may be of interest include:

Roland P (1981) Somatotopical tuning of postcentral gyrus during focal attention in man. A regional cerebral blood flow study. Journal of Neurophysiology 46 (4):744-754

Johansen-Berg H, Christensen V, Woolrich M, Matthews PM (2000) Attention to touch modulates activity in both primary and secondary somatosensory areas. Neuroreport 11 (6):1237-1241

Hamalainen H, Hiltunen J, Titievskaja I (2000) fMRI activations of SI and SII cortices during tactile stimulation depend on attention. Neuroreport 11 (8):1673-1676. doi:10.1097/00001756-200006050-00016

Puckett AM, Bollmann S, Barth M, Cunnington R (2017) Measuring the effects of attention to individual fingertips in somatosensory cortex using ultra-high field (7T) fMRI. Neuroimage 161:179-187. doi:10.1016/j.neuroimage.2017.08.014

Yu Y, Huber L, Yang J, Jangraw DC, Handwerker DA, Molfese PJ, Chen G, Ejima Y, Wu J, Bandettini PA (2019) Layer-specific activation of sensory input and predictive feedback in the human primary somatosensory cortex. Sci Adv 5 (5):eaav9053. doi:10.1126/sciadv.aav9053

More useful references could be found in this recent review:

https://doi.org/10.1016/j.neuroimage.2020.117255

2) The Materials and methods would benefit from additional rationale / supporting references throughout. Whereas it is generally clear what was done, it is sometimes less clear why certain choices were made. Perhaps some of the choices are "standard practice" when working with single unit recordings, but I was left in want of a bit more reasoning (or at least direction to relevant literature). Some examples are below:

For the population correlation: why was the correlation computed 250 times or why were the two distributions shuffled together 2000 times?

For the decode analysis: consider providing a reference for those interested in better understanding the "peeking" effects mentioned.

Response latency: how were window parameters determined (for both visualization and the latency calculation). And what was the rationale for them being different – especially given that the data used for the response latency calculation was still visualized (at least in part)? Relatedly, I'd be curious to see the entire time-course for that data rather than just the shaded region of the "visualization" data. Also, it would be nice if a comment (or some data) could be provided regarding how much the latency estimates change based on these parameter choices.

Temporal dynamics of population activity: why use a 500 ms window, stepped at 100 ms intervals instead of something else?

Temporal dynamics of single unit activity: it is stated that the neurons were restricted to those whose 90th percentile accuracy was at least 50% to ensure only neurons with some degree of significant selectivity were used for the cluster analysis. But why these particular values? Are the results sensitive to this choice? In this section, I'd also suggest providing references for those interested in better understanding the use of Bayesian information criteria. Similarly, it is stated that PCA is a "standard method for describing the behavior of neural populations" – as such it would be nice to provide some relevant references for the reader

3) At the start of the Results section it is stated that the recordings were from "well-isolate and multi-unit neurons". This seems to contradict the Materials and methods section, which only talks about "sorted" neurons. This needs to be clarified, and if multi-units were included, it should be stated which sections this concerns as it will have implications for the results (e.g. for selectivity for different body parts). In any case, the number of neurons included in different analyses should be evident. There are some numbers in the Materials and methods and sprinkled throughout the Results section, but for some of the analyses (e.g. clustering analysis, which was run only on a responsive subset of neurons) no numbers are provided.

4) The linear analysis section needs further details. The coefficients are matched to "conditions" but it is not explained how. I am assuming that each touch location is assigned to a condition c, however the way the model is described suggests that the vector X can in principle have multiple conditions active at the same time. Additionally, could the authors confirm whether it is the significance of the coefficients that determined whether a neuron was classed as responsive as shown in Figure 1? This analysis states a p-value but does give no further information on which test was run and on what data.

5) Figure 1C could be converted into a matrix that lists all combinations of RF numbers on either side of the body to highlight whether larger RFs on one side of the body generally imply larger RFs on the other side.

6) I am confused about the interpretation of the coefficient of determination as shown in Figure 2A. In the text this is described as testing the "selectivity" of the neurons. To clarify, I am assuming that the "regression analysis" is referring to the linear model described in a previous section. The authors then presumably took the coefficients from this model for a single side only and tested how well they could predict the responses to the opposite side, as assessed by R^2^ (Figure 2C,E). Before that in Figure 2A, they tested how well each single-side model could predict the responses. This is all fine, but the "within" comparison then simply tests how well a linear model can explain the observed responses, and has nothing to do with the selectivity of the neuron. For example, the neuron might be narrowly or broadly selective, but the model might fit equally well.

7) We computed a cross-validated coefficient of determination (R^2^ within) to measure the strength of neuronal selectivity for each body side.

Even after reading the methods (further comments below) it is difficult to figure out what all these related measures reveal. At this point in the text it is very difficult to intuit how R^2^ would measure selectivity.

8) Regarding the timing analysis, it is not clear to me how the accuracy can top out at 100% as shown in the figure, when the control conditions were included. Additionally, the authors should state the p value and statistic for the comparison of latencies.

9) Spatial analysis. Could the authors provide the size of the paintbrush tip that was used in this analysis. Furthermore, as stimulation sites were 2 cm apart, it is not appropriate to specify receptive fields down to millimeter precision.

10) Imagery: how many neurons were responsive to both imagery and real touch? Were all neurons that were responsive to imagery also responsive to actual touch? This is left vague and Figure 5 only includes the percentages per condition, but gives no indication of how many neurons responded to several conditions. Whether and how many neurons were responsive to both conditions also determines the ceiling for the correlation analysis in Figure 5D (e.g. if the most neurons are responsive only to actual but not imaginary touch, this will limit the population correlation).

11) What is added by including both classification and Mahalanobis distance?

Relatedly, information coding evolves for a single unit. Two complimentary analyses were then performed.

In what sense are they complementary? What is added (besides complexity) by including both cluster analysis and PCA?

12) Classification was performed using linear discriminant analysis with the following assumptions:

one, the prior probability across tested task epochs was uniform;

It is not clear what prior probability this refers to. Just stimulus site?

two, the conditional probability distribution of each unit on any epoch was normal;

Is this a reference to firing rate probability conditioned on stimulus site?

three, only the mean firing rates differ for unit activity during each epoch (covariance of the normal distributions are the same for each);

four, firing rates for each input are independent (covariance of the normal distribution is diagonal).

Does this refer to independent firing rates of neurons across stimulus sites? This seem very unlikely, given everything we know about dimensionality of cortex. Perhaps it refers to something else. Cannot all of these assumptions be tested? Were they?

13) We computed the cross-validated coefficient of determination

(R2 within) to measure how well a neuron's firing rate could be explained by the responses to the sensory fields.

This needs a better description, and I may be missing the point entirely. I assume it is an analysis of mean firing rate (which should be stated explicitly) and that it uses something like the indicator variable of the linear analysis of individual neuron tuning above. In this case is this is a logistic regression? As it is computed for each side independently, it would appear that there are only four bits to describe the firing of any given neuron. This would seem to be a pretty impoverished statistic, even if the statistical model is accurate.

14) The purpose of computing a specificity index was to quantify the degree to which a neuron was tuned to represent information pertaining to one side of the body over the other.

This is all pretty hard to follow. The R2 metric itself is a bit mysterious, as noted above. Within and across R2 is fairly straightforward, but adds to the complexity, as does SI, which makes comparisons of three different combinations of these measures across sides. Aside from R2 itself, the math is pretty transparent. However, a better high-level description of what insight all the different combinations provide would help to justify using them all. As is, there is no discussion and virtually no description of the difference across these three scatter plot. The critical point apparently, is that, "nearly all recorded PC-IP neurons demonstrate bilateral coding". There should be much a more direct way to make this point.

15) Computing response latency via RF discrimination is rather indirect and assumes that there is significant classification in the first place. I suspect it will add at least some delay beyond more typical tests. Why not a far simpler and more direct test of means in the same sliding window? Alternatively, a change point analysis?

[Editors' note: further revisions were suggested prior to acceptance, as described below.]

Thank you for submitting your article "Neural encoding of felt and imagined touch within human posterior parietal cortex" for consideration by *eLife*. Your article has been reviewed by two peer reviewers, and the evaluation has been overseen by Tamar Makin as the Senior and Reviewing Editor. The reviewers have opted to remain anonymous.

The reviewers have discussed the reviews with one another and the Reviewing Editor has drafted this decision to help you prepare a revised submission.

The reviewers were happy with the vast majority of the extensive revisions that have been implemented relative to the original submission, but felt that the presentation of the results and their discussion was inadequate. While we recognise the enormous amount of work that was poured into the revisions, there are still a few issues, originating from the revisions, that the reviewers felt require further consideration. But beyond these specific considerations that are elaborated below, we all agreed that the paper, at its present form, is too dense with results/interpretation, making it very difficult to read and evaluate. As one of the reviewers summarised:

"My main high-level concern was the, "extensive and overlapping analyses" that made it difficult to follow and to find a clear takeaway message from the paper. I suggested eliminating a number of figures and clarifying the remainder to improve the impact of the paper. Although a few small panels have disappeared, most of the duplication remains and there are now nine, fairly complex figures (had been eight). It's hard to judge how much longer the text got because the figures are no longer embedded, but that has certainly also increased. The paper has not improved as a consequence".

There is a strong need for a “deep clean” (or more precisely deep edit) of the paper. As highlighted in our original review, there's a lot of overlap between analyses/findings that will not be of interest to the average reader. We believe the key innovation of the study – the imagery results – could be presented a lot more concisely with a more focused discussion of the findings and the limitations of the suggested interpretation. Much of the details of the tactile RF properties is of secondary interest and as such should be moved to the supplementary section. I leave the decision of how to achieve this leaner and more focused version to the authors, but please consider the average reader (and their very limited time and attention span) when making the edits. As a rule of thumb, the reviewers estimated you will be likely be able to remove ~50% of the figure panels and 33% of the main text without weakening the key findings.

1) Perhaps the most exciting innovation of the study relates to the neural responses related to the imagery conditions. Yet, we know from many previous studies in humans that cognitive processes…

While the Introduction has been much improved, the Discussion still mostly disregards the alternative cognitive processes that are likely to drive the present findings (prediction, attention). The authors seem to downplays the impact of attention considerably. For example, they state:

"Most studies of pre-stimulus attention report that any modulation of baseline activity is modest at best."

But especially when considering the modest effects of imagery, this statement is misleading. The Roland (1981) study that we provided, for example, show a 25% increase in rCBF when subjects attended the index finger without being stimulated. And this increase was spatially specific as it shifted to the lip area when attending to the upper lip instead of the finger. Although the other 3 attention references were looking at the effects of attention while sensory stimulation was present, they still seem relevant to the discussion and show what I'd consider to be greater than modest effects of attention. For example, the Puckett (2017) reference shows clear digit-specific modulation when attending to the different fingertips. Note that despite stimulation being present during this condition, the stimulation was constant (allowing the phasic attention-related signal to be somewhat isolated from the sensory signal). The amplitude of the phasic attention signal was similar to that elicited by phasic stimulation alone (i.e., without endogenous attention).

There needs to be a more serious consideration that the effects attributed here to imagery are in fact modulated by (or even driven by) related cognitive processes, such as prediction and attention, which are not specifically linked to the auditory cue used in the present study.

2) Figure 4: Several panels would be more effective…

This figure has changed quite lot, addressing my cosmetic concerns. However, I do not understand this statistical test: If no comparison was significant (FDR corrected), the unit was classified as "single peak." If at least one of the comparisons was statistically significant, it was characterized as "multi-peak." I must be missing something fundamental. I took this to be a test of differences of the responses to the different body parts, with respect to the peak response. How is this a single peak? No differences sounds like a flat line. By this definition a neuron with no response whatsoever would be "single peak". Likewise, the multi-peak definition is a puzzle.

3) What is added by including both classification and Mahalanobis distance?

"Mahalanobis distance that provides a sensitive measure of change which is masked by the discretization process of classification "

I don't think the discrete nature of the classifier output is really the biggest issue. By averaging across many instances, it essentially becomes continuous, as in this figure. There are also classifier-related metrics that are by their nature, continuous. The nature of the distance measure they are making is likely more important. In this case, I do not understand why the classification rises only slightly at t=0 as the M distance increases sharply. Subsequently, between 1 and 2.5 s, classifier success drops back below its original level even as distance is stable. The two measures really don't seem to be concordant. What is going on here? I think this concern is not unrelated to my next comment about Figure 8C (now 9C).

4) Figure 8C: Despite my best efforts, I have no idea…

"This asymmetry is likely a consequence of the analysis technique and may not be of physiological significance.”

I agree with the statement, but not its sentiment. Perhaps I'm missing something, but the fact that a single classifier can distinguish between rest state and two very different activation states does nothing to suggest those two states are a general representation of an input. The classification failure in the opposite direction only reinforces that. Presumably, classifying imagined and actual touch would be trivial, at a much higher level of success than rest and imagined touch, suggesting that they are in fact, rather different, even by this metric. If the authors wish to make the claim that their results show more than grossly common receptive fields bilaterally and across modes (which is not an uninteresting finding) they would do well to adopt tools more appropriate for it, like those that have been used by the groups of Shenoy, Churchland, Miller, Kaufman, and others: Canonical correlations, principal angles, subspace overlap…

5) Computing response latency via RF discrimination is rather indirect…

The authors have adopted a more sensible and sensitive test of latency. I do not agree with this statement, however: "We believe this is likely related to discussion above about Mahalanobis distance versus classification: namely, changes in the underlying neural behavior are only detected once the neural responses cross a decision line which likely results in delays detecting changes in neural behavior." In what fundamental sense is classification significance different from a significant distance in M-space? It seems to me that the more likely explanation is simply that significant modulation precedes significant discrimination.

---

## [Author Response]

Overall, we found the manuscript to be well-written, the study to be interesting, and for the most part the analyses are well thought out. But at the same time, the reviewers raised multiple main concerns regarding missing information and unclear descriptions of some of the analyses undertaken, which are detailed below. In addition, it was felt that there was unnecessary overlap across analyses – the first part especially contains a number of analyses that seem to make very similar points repeatedly or where it is not entirely clear what the point is in the first place. As such, there is a need to identify and cut a lot of the duplicative analyses/results and explain both the essential methods and the interpretation of the remaining results more succinctly and clearly. The key analyses could then be streamlined and better justified, ideally with an eye towards a consistent approach in both parts of the paper. As you will see, most of the comments detailed below are geared towards guiding the authors through this major revision (please forgive any duplicative comments on our behalf, given the technical nature of most of the comments, the Editor was keen to preserve the original comments as made by the reviewers). In addition, there are also some major considerations regarding the contextualisation and interpretation of the key imagery results, as detailed in the first major comment below.1) Perhaps the most exciting innovation of the study relates to the neural responses related to the imagery conditions. Yet, we know from many previous studies in humans that cognitive processes (most notably attention), can modulate somatosensory responses. What the current study offers is an opportunity to characterise this cognitive modulation at a single neuron (and population) level over space and time, with hopes of achieving better insight into its underlying mechanism. While the reviewers agree that this unique contribution is valuable, it was also agreed that the manuscript would benefit from additional context in the Introduction as well as a more thorough discussion – particularly with respect to related research on this topic and potential mechanisms that could be recruited to support the observed neural modulation during the imagery condition.Introduction: The second paragraph did well in establishing why one might be interested in examining somatosensory processing in the PPC. It was however, less clear why the particular questions at the end of the paragraph were being posed. Perhaps an extra paragraph could be added to bridge the notion that a sizeable body of literature has been developed around somatosensory representation within the PPC and the several "fundamental" questions remaining that are of interest here.

We have expanded the Introduction to better situate the questions we address in this manuscript. We have tried to highlight that despite a growing body of literature surrounding higher-order somatosensory processing within the PPC (both in humans and in animal models), there exist gaps in our knowledge related to the spatial structure of tactile receptive fields and whether tactile cognition engages the same populations of cells with similar spatial tuning preferences.

The modified text is below.

Introduction:

“Touch is a complex, multisensory perceptual process (1-3). In non-human primates (NHPs), multisensory input (e.g., visual, tactile) converges upon neurons in higher-order brain regions such as the posterior parietal cortex (PPC) where they are integrated into coherent representations (3-11). […] The latter represents a novel finding, thus far untestable in NHP models, and suggests PPC involvement in the cognitive processing of touch.”

Discussion: The manuscript would benefit from a more thorough discussion of "imagination per se" and the various top-down processes that might be involved – as well as better positioning with respect to previous studies investigating top-down modulation of the somatosensory system. The authors state that the cognitive engagement during the tactile imagery may reflect semantic processing, sensory anticipation, and imagined touch per se – which we would not argue. But we would also expect some explicit mention of processes like attention and prediction. Perhaps these are intended to be captured by "sensory anticipation" – but, for example, attention can be deployed even if no sensation is anticipated. Importantly, it seems that imagining a sensation at a particular body site might well involve attending to that body part. That is, one may first attend to a body part before "imagining" a sensation there – and then even continue to attend there the entire time the imagining is being done. Because of this, perhaps the authors are considering attention to be a part of "imagination per se". But since attention has been shown to modulate somatosensory cortex without imagination, how can one exclude the possibility that the neuronal activity measured here simply reflects this attention component?Regardless, we think the Discussion would benefit from a more explicit treatment of these top-down processes – especially given the number of previous studies showing that they are able to modulate activity throughout the somatosensory system. Some literature that may be of interest include:Roland P (1981) Somatotopical tuning of postcentral gyrus during focal attention in man. A regional cerebral blood flow study. Journal of Neurophysiology 46 (4):744-754Johansen-Berg H, Christensen V, Woolrich M, Matthews PM (2000) Attention to touch modulates activity in both primary and secondary somatosensory areas. Neuroreport 11 (6):1237-1241Hamalainen H, Hiltunen J, Titievskaja I (2000) fMRI activations of SI and SII cortices during tactile stimulation depend on attention. Neuroreport 11 (8):1673-1676. doi:10.1097/00001756-200006050-00016Puckett AM, Bollmann S, Barth M, Cunnington R (2017) Measuring the effects of attention to individual fingertips in somatosensory cortex using ultra-high field (7T) fMRI. Neuroimage 161:179-187. doi:10.1016/j.neuroimage.2017.08.014Yu Y, Huber L, Yang J, Jangraw DC, Handwerker DA, Molfese PJ, Chen G, Ejima Y, Wu J, Bandettini PA (2019) Layer-specific activation of sensory input and predictive feedback in the human primary somatosensory cortex. Sci Adv 5 (5):eaav9053. doi:10.1126/sciadv.aav9053More useful references could be found in this recent review:https://doi.org/10.1016/j.neuroimage.2020.117255

We significantly modified the Discussion to better discuss possible contributions to cognitive touch processing, especially imagery.

The pertinent modified text from the Discussion is below.

“What does neural processing within human PC-IP during tactile imagery represent?

While our task identifies dynamic engagement of multiple cognitive processes during tactile imagery, it is inadequate to precisely define the cognitive correlates of the observed neural activity. […] The above cognitive phenomena may each independently engage the same neural population as distinct phenomena, or may be distinct processes that nonetheless engage the same underlying neural substrate.”

2) The Materials and methods would benefit from additional rationale / supporting references throughout. Whereas it is generally clear what was done, it is sometimes less clear why certain choices were made. Perhaps some of the choices are "standard practice" when working with single unit recordings, but I was left in want of a bit more reasoning (or at least direction to relevant literature).

We have updated the Materials and methods section to clarify the analyses, provide additional technical details, expand motivation for analysis choices, and to point the reader to supporting references for further information.

Some examples are below:For the population correlation: why was the correlation computed 250 times or why were the two distributions shuffled together 2000 times?

In the correlation analysis, for each condition (touch location), we split the data into test and training sets, averaged responses across repetitions, and finally computed a correlation to measure population response similarity within and across conditions. Because the correlation was computed for independent training and testing sets, within and across condition correlations can be directly compared. We repeated this process 250 times, each time creating new random assignments of which trials are put into the training and test sets. Each split gives a slightly different correlation value, and we average across all values to give our best estimate of the actual population correlation. Performing the splits 250 times is somewhat arbitrary, but was chosen based on an empirical analysis applied to preliminary data. For preliminary data, we performed the analysis with N splits, with N ranging from 5 to 200 in steps of 5. We found that the mean correlation across splits converged to a stable value by about 80 splits. We then roughly tripled that number to be safe. So 250 was chosen to ensure that the numerical sampling scheme would capture a stable value of our cross-validated correlation metric. A similar process was used for performing 2000 shuffles as part of our permutation shuffle test.

The following was added to the relevant Materials and methods section (“Population Correlation”)

“Performing the splits 250 times was chosen based on an empirical analysis applied to preliminary data. For the preliminary data, we performed the analysis with N splits, with N ranging from 5 to 200 in steps of 5. We found that the mean correlation across splits converged to a stable value by approximately 80 splits. We then roughly tripled this value to ensure that the numerical sampling scheme would capture a stable value of our cross-validated correlation metric.”

“As in the case above, our permutation shuffle test used 2000 shuffles to ensure that the numerical sampling scheme would capture a stable value of the percentile of our true difference as compared to the empirical null distribution.”

For the decode analysis: consider providing a reference for those interested in better understanding the "peeking" effects mentioned.

The “peeking” phenomena refers to overestimation of generalization error when using supervised feature selection on the entire dataset. We have added a reference to the text, Smialowski, Frishman and Kramer, 2010. To quickly touch on the subject and provide intuition, imagine that you have a population of 1000 neurons that are unmodulated by task condition. By chance, some of these units may appear task-modulated: with a p-value of 0.05, we would expect to find 50 or so “modulated” units. This is expected to occur by chance and is the reason multiple comparisons corrections are needed. Here, peeking refers to the situation where neurons are preselected to be modulated prior to cross-validated classification analysis. By cherry-picking “modulated” units you break the logic of cross-validation which requires independence between training and testing sets.

Response latency: how were window parameters determined (for both visualization and the latency calculation). And what was the rationale for them being different – especially given that the data used for the response latency calculation was still visualized (at least in part)? Relatedly, I'd be curious to see the entire time-course for that data rather than just the shaded region of the "visualization" data. Also, it would be nice if a comment (or some data) could be provided regarding how much the latency estimates change based on these parameter choices.

We are replacing the original version of the latency analysis. In line with other reviewer comments, in the updated manuscript, we measure latency using the basic responsiveness, not discriminability, of the neural population. One benefit of the updated approach is that we did not have to introduce any additional smoothing beyond averaging activity within 2 ms windows. The updates and rationale for the new latency calculation are discussed in response to major point 15.

The following is not included in the revised manuscript as we have revised how we perform the latency analysis. While no longer pertinent to the latency analysis in the revised manuscript, we include a response below for completeness:

As background, selecting the window size/smoothing kernel involves a balance between preserving temporal resolution (which asks for a smaller time window) and mitigating against noise (which asks for a larger window). For computing the latency estimate, fine temporal resolution is critical and so we chose a narrow smoothing kernel, essentially as small as we could make it without high-frequency noise obscuring the transition. Given our truncated Gaussian kernel, increasing the kernel width (smoothing more) pulls in information from the future and past into the firing rate estimate at the current moment in time. In computing latencies, this is highly problematic as it destroys any ability to resolve the absolute timing of signals. With our truncated Gaussian kernel, increasing the smoothing kernel for computing latency simply results in earlier and earlier latency estimates, but only because the firing rate estimate is allowed to peak further and further into the future. For the “visualization” kernel, the latency was actually pre-stimulus (~-100 ms). It is possible to use “causal” filters that can only smooth by averaging past data (not past and future.) This does not solve the core problem though. Increasing the smoothing width in this case simply delays the latency more and more as an increasing amount of the data used to compute the estimate comes from pre-stimulus activity thus adding noise and decreasing sensitivity. This same basic challenge is inherent in all techniques, although it is not always explicit. For example, hidden Markov models essentially use exponential smoothing where the smoothing coefficient is indirectly set by the window durations used for the pre and post-stimulus phases (which is still an experimenter choice).

Please see Author response image 1 for a plot of the full window with the “latency” smoothing kernel. Orange=contralateral, blue = ipsilateral. It’s fairly innocuous and essentially shows the same results as the “visualization” kernel with more high-frequency noise (as expected).

**Author response image 1. sa2fig1:** 

We understand why choosing two separate smoothing’s, especially without better justification, warranted scrutiny. In brief, the only point of the “visualization” was to show that classification performance over the course of the trial was well behaved, with chance performance before the stimulus and sustained accurate performance after. For this basic point, a smoother signal with larger time steps seemed appropriate and required less time to compute. Again, the new version uses no smoothing and only has one visualization (see point 15).

Temporal dynamics of population activity: why use a 500 ms window, stepped at 100 ms intervals instead of something else?

Unlike the original latency analysis, for all other analysis we used a 500 ms window size for averaging time series data (also called box-car smoothing). We chose a fixed window size (as opposed to e.g. truncated Gaussian) as the exact bounds of what data was used in analysis is transparent for any particular analysis window. The use of 500 ms is somewhat arbitrary and was chosen as a balance between temporal resolution and noise mitigation. While the exact choice of window size can massage around various metrics (e.g. larger smoothing window can increase R^2^), for anything with decent signal-to-noise, as is the case with our data, the exact choice of smoothing size is fairly inconsequential so long as the same kernel is used when making comparisons between different conditions. The choice of step size is essentially anchored to the choice of smoothing window. We could have stepped at 1ms, but with a 500 ms smoothing window, such a small step is not justified. 100 ms, representing a change in 20% of the data, allows us the ability to temporally localize changes in neural responses while being efficient with our compute time. We have added text to the Materials and methods (“Temporal dynamics of population activity during tactile imagery task: within category”) as well to clarify the choice of time window and step size. The text is copied below.

“We used a fixed window size for averaging time series data for analysis (box-car smoothing) as it provides straight-forward bounds for the temporal range of data that are included in the analysis for a particular time window. 500 ms was chosen as a good balance between temporal resolution and noise mitigation. We note that although the window size can influence various metrics (e.g., larger smoothing windows can increase coefficients of determination, R^2^) the choice of smoothing size is largely inconsequential as long as the kernel size is kept consistent when making comparisons across conditions. The choice of a 100 ms step size was anchored to the choice of smoothing window. A small step, such as 1 ms would not be justified with a 500 ms time window. We chose 100 ms, representing a change in 20% of the data, to allow us the ability to temporally localize changes in neural response without unnecessarily oversampling a smoothed signal and thus not unnecessarily increasing computation time for analysis.”

Temporal dynamics of single unit activity: it is stated that the neurons were restricted to those whose 90th percentile accuracy was at least 50% to ensure only neurons with some degree of significant selectivity were used for the cluster analysis. But why these particular values? Are the results sensitive to this choice? In this section, I'd also suggest providing references for those interested in better understanding the use of Bayesian information criteria. Similarly, it is stated that PCA is a "standard method for describing the behavior of neural populations" – as such it would be nice to provide some relevant references for the reader

In consideration of the reviewers’ comments, and for simplicity of presentation, we have eliminated the cluster analysis previously presented in Figure 7A. Figure 7B (the PCA analysis) has been moved to Figure 8D. We opted to go with the PCA approach as PCA is more commonly used in the literature and involves fewer assumptions. We have included a reference to the use of PCA in neuroscience, below.

Cunningham JP, Yu BM. Dimensionality reduction for large-scale neural recordings. Nat Neurosci. 2014;17(11):1500-9.

While no longer in the manuscript, we briefly explain the choices made for the cluster analysis for completeness. We wanted to ensure that we are applying the classification analysis only to units with a reasonable strength of encoding. The cross-validated classification analysis generates a large matrix of values that, by chance, can sometimes result in large values. We are interested in the maximum classification accuracy but wanted to avoid these chance peaks. We therefore selected the 90^th^ percentile to provide a more robust estimate of the peak value. The 50% accuracy number was chosen through a shuffle analysis, in which we found that it reflected greater than the 95^th^ percentile of chance outcomes. We performed a basic sensitivity analysis, testing the 80^th^, 85^th^ and 90^th^ percentile, and threshold values of 55%, 55% and 60% percent. The basic result, with three clusters and relatively similar temporal patterns, were robust to these choices.

For further reading on Bayesian information criteria:

Original article:

G. E. Schwarz, Estimating the Dimension of a Model. *Annals of Statistics***6**, 461–464 (1978).

For use in clustering, see:

S. S. Chen, P. S. Gopalakrishnan, Clustering via the Bayesian information criterion with applications in speech recognition. *Proceedings of the 1998 IEEE International Conference on Acoustics, Speech and Signal Processing***2**, 645-648 (1998).

3) At the start of the Results section it is stated that the recordings were from "well-isolate and multi-unit neurons". This seems to contradict the Materials and methods section, which only talks about "sorted" neurons. This needs to be clarified, and if multi-units were included, it should be stated which sections this concerns as it will have implications for the results (e.g. for selectivity for different body parts). In any case, the number of neurons included in different analyses should be evident. There are some numbers in the Materials and methods and sprinkled throughout the Results section, but for some of the analyses (e.g. clustering analysis, which was run only on a responsive subset of neurons) no numbers are provided.

There is no contradiction, but we understand the confusion. All channels were sorted into putative neurons, which we then categorized as either being single unit or multi-unit neurons. Multi-unit refers to the situation where two or possibly more neurons have sufficiently similar waveforms that they cannot be distinguished. All units were pooled for the analyses presented in the manuscript. To ensure that the pooling did not impact the validity of core results, we performed separate analyses on the single and multi-unit responses based on the cluster “isolation distance” computed during the spike sorting stage and found that the basic conclusions were consistent across single and multi-unit data. We have now included new supplemental figures (Figure 1—figure supplement 2, Figure 2—figure supplement 2, Figure 3—figure supplement 2, Figure 4—figure supplement 1 and Figure 7—figure supplement 1) that show the results for the pooled, single unit, and multi-unit data for representative analyses that may be especially sensitive to mixing response properties from multiple units. As can be seen, the qualitative results are similar for multi and single-unit data. We have included within the Materials and methods (“Data collection and unit selection”) a clarification on unit selection, pooling, the invariance of our results to the pooling of units, and our approach to categorizing units as single units or potentially multi-units.

Materials and methods:

“Well isolated single and multi-units were pooled across recording sessions. To ensure that such pooling did not bias the conclusions of this manuscript, we performed core analyses within this manuscript on single units alone, potential multi-units alone and all units together. The results of these analyses, shown as supplemental figures for key results, generally demonstrate that our results were robust to the pooling of all sorted units together. For the separation of spike sorted units into high quality single and multi-units, we used as a threshold the mean across all units of the base-10 logarithm of their cluster isolation distances, based on a previously described method (18, 50). Sorted units for which the cluster isolation distance was above this measure were considered single units, and those with a distance below this threshold were considered potential multi-units. Findings were robust to the exact choice of isolation distance threshold.”

4) The linear analysis section needs further details. The coefficients are matched to "conditions" but it is not explained how. I am assuming that each touch location is assigned to a condition c, however the way the model is described suggests that the vector X can in principle have multiple conditions active at the same time. Additionally, could the authors confirm whether it is the significance of the coefficients that determined whether a neuron was classed as responsive as shown in Figure 1? This analysis states a p-value but does give no further information on which test was run and on what data.

Your assumptions are correct. The description of the linear analysis (Materials and methods, “Linear analysis” relevant for Figures 1, 5 and 7) has been updated to improve clarity. The relevant text is included below:

“To determine whether a neuron was tuned (i.e., differentially modulated to touch locations), we fit a linear model that describes firing rate as a function of the neuron’s response to each touch location. […] A unit was considered tuned if the F-statistic comparing the β coefficients was significant (*p*<0.05, false discovery rate (FDR) corrected for multiple comparisons).”

5) Figure 1C could be converted into a matrix that lists all combinations of RF numbers on either side of the body to highlight whether larger RFs on one side of the body generally imply larger RFs on the other side.

We have converted Figure 1C (currently Figure 2B) to matrix form. We agree that seeing that larger RFs on one side imply larger RFs on the other side is a nice feature of this presentation.

6) I am confused about the interpretation of the coefficient of determination as shown in Figure 2A. In the text this is described as testing the "selectivity" of the neurons. To clarify, I am assuming that the "regression analysis" is referring to the linear model described in a previous section. The authors then presumably took the coefficients from this model for a single side only and tested how well they could predict the responses to the opposite side, as assessed by R^2^ (Figure 2C,E). Before that in Figure 2A, they tested how well each single-side model could predict the responses. This is all fine, but the "within" comparison then simply tests how well a linear model can explain the observed responses, and has nothing to do with the selectivity of the neuron. For example, the neuron might be narrowly or broadly selective, but the model might fit equally well.

I think your core understanding is correct; but our usage of “selectivity” created unnecessary confusion. We understand the confusion and have taken several steps to clarify. First, we were using selectivity here to refer to the ability to accurately differentiate between the different touch locations using the linear model. You are correct that this would have nothing to do with narrow or broad tuning. To avoid this confusion, we have replaced “selective” with “discriminative” throughout the manuscript. Further, we have clarified the text describing what is to be learned from this analysis, including an illustrative schematic, Figure 3—figure supplement 4. The relevant updates (from Materials and methods: “Tests for mirror symmetric neural coding of body locations: single unit analysis” (relevant for Figure 3) are included below:

“The purpose of this analysis was to assess whether neural responses to one body side were the same as neural responses to the alternate body side on a single unit basis. Heuristically, we used a cross-validation approach, similar in concept to the population correlation, to ask whether the neural responses to one body side are similar to the other body side. […] If the patterns of response are similar, this would lead to data points falling along the identity line. If the patterns are distinct, the data points should fall below the identity line.”

7) We computed a cross-validated coefficient of determination (R^2^ within) to measure the strength of neuronal selectivity for each body side.Even after reading the Materials and methods (further comments below) it is difficult to figure out what all these related measures reveal. At this point in the text it is very difficult to intuit how R^2^ would measure selectivity.

The confusion here is very much related to point 6. Again, we believe we understand the source of the confusion and have replaced “selective” with “discriminative” and have updated the relevant text for clarity (see response to major point 6).

8) Regarding the timing analysis, it is not clear to me how the accuracy can top out at 100% as shown in the figure, when the control conditions were included. Additionally, the authors should state the p value and statistic for the comparison of latencies.

Based on reviewer feedback and additional analysis, we have changed the way latency is computed (see major point 15). In the original manuscript, only the experienced touch conditions were included in the sliding window classification analysis: touch to the shoulder, cheek, and hand. With the three conditions, chance performance is 33% and accuracy can hit 100%, consistent with the previous results.

A detailed description of the new approach includes a description of how the p-value was computed based on a permutation shuffle test (see major point 15).

9) Spatial analysis. Could the authors provide the size of the paintbrush tip that was used in this analysis. Furthermore, as stimulation sites were 2 cm apart, it is not appropriate to specify receptive fields down to millimeter precision.

The Materials and methods have been updated to include the size of the paintbrush tip (3 mm).

Materials and methods have been updated to read:

“Stimuli were presented via a paintbrush (three mm round tip) gently brushed against each location, at a frequency of one brush per second.”

The millimeter place has been removed.

10) Imagery: how many neurons were responsive to both imagery and real touch? Were all neurons that were responsive to imagery also responsive to actual touch? This is left vague and Figure 5 only includes the percentages per condition, but gives no indication of how many neurons responded to several conditions. Whether and how many neurons were responsive to both conditions also determines the ceiling for the correlation analysis in Figure 5D (e.g. if the most neurons are responsive only to actual but not imaginary touch, this will limit the population correlation).

We now include in Figure 7 (panel D) a Venn diagram that illustrates the number of units that were significantly modulated by experienced touch alone, experienced touch and imagery, and imagery alone. The analysis was restricted to the cheek and shoulder given that touch induced no response on the hand.

11) What is added by including both classification and Mahalanobis distance?

We used classification analysis as the values are readily interpretable and it captures the key feature that we think is important: the ability to discriminate between the different task conditions and whether the decision boundaries generalize between experimental manipulations, e.g. between imagery and experienced touch. However, the discretization process in classification analysis can lead to a loss in sensitivity. The population response might be changing, but not in a way that crosses the decision boundaries established by the classification algorithm. Mahalanobis distance is very sensitive to changes in neural population response. A limitation of Mahalanobis distance is that we, for example, cannot tell whether the changes will affect how e.g., a classifier would interpret the population response. Thinking about this in the context of Figure 8C (previously Figure 6C), the classification analysis tells us that the basic population level response patterns that allow us to distinguish the different classes are generally preserved from the cue through the execution period. The Mahalanobis adds to this picture by showing that there is a clear transition from one point to another in the neural state space, and that this transition occurs at short latency and over a short duration just after the go cue.

A description motivating the benefit of Mahalanobis distance is included in the Materials and methods (“Temporal dynamics of population activity during tactile imagery task: within category” (relevant for Figure 8)):

“…to explicitly test whether population activity was changing, we used Mahalanobis distance as our measure. This is necessary as classification involves a discretization step that makes the technique relatively insensitive to changes in neural population activity that do not cross decision thresholds. Mahalanobis distance, being a proper distance measure, is a more sensitive measure of change. To illustrate, imagine that a classifier is trained on time point A and tested on time point B. At time point A, the means of the two classes are 0 and 1 respectively and at time point 2 the means are 0 and 4 respectively. All classes are assumed to have equal but negligible variance (e.g. 0.01) in this example. When trained on time point A, the classifier finds a decision boundary at 0.5. with 100% classification accuracy. When tested on time point B, with the same 0.5 decision boundary, the classifier again is 100%. Naively, this could be interpreted as signifying that no change in the underlying data has occurred, even though the mean of the second distribution has shifted.”

Relatedly, information coding evolves for a single unit. Two complimentary analyses were then performed.In what sense are they complementary? What is added (besides complexity) by including both cluster analysis and PCA?

They are complimentary in the sense that while PCA estimates how much of the variability across all units is captured by specific temporal patterns while clustering can show how many units are described by the different patterns. However, aggregating across reviewer responses, and acknowledging that the primary message from both is essentially the same, we have opted to simplify presentation and are removing the cluster analysis.

12) Classification was performed using linear discriminant analysis with the following assumptions:one, the prior probability across tested task epochs was uniform;It is not clear what prior probability this refers to. Just stimulus site?two, the conditional probability distribution of each unit on any epoch was normal;Is this a reference to firing rate probability conditioned on stimulus site?three, only the mean firing rates differ for unit activity during each epoch (covariance of the normal distributions are the same for each);four, firing rates for each input are independent (covariance of the normal distribution is diagonal).Does this refer to independent firing rates of neurons across stimulus sites? This seem very unlikely, given everything we know about dimensionality of cortex. Perhaps it refers to something else. Cannot all of these assumptions be tested? Were they?

These assumptions are more about tractability of analysis given limited data than true assumptions about the behavior of neurons. To elaborate, in linear discriminant analysis, the response to each stimulation site is modeled as a Gaussian characterized by a mean and variance. In our experiments, we only acquire 10 repetitions per condition, generally not enough to get a reasonable estimate of variance. Therefore, we compute a pooled estimate of variance based on all conditions to try and get a better variance estimate. Likewise, estimates of covariance require even more data. We sometimes use the Ledoit-Wolf optimal shrinkage estimator for covariance to compute a regularized estimate, but this almost invariably returns the diagonal matrix given our amount of data and thus it feels somewhat disingenuous to claim we are capturing covariance structure between neurons. To validate these choices, we use cross-validation to test how well the models explain held-out data. For on the order of ~10 repetitions per condition, the assumptions above nearly always outperform or at worst match performance for alternative assumptions. As an aside, we have collected data in other experiments where we have on the order of ~80 trials per condition. With this amount of data, allowing separate estimates of variance and a full covariance matrix (sometimes called quadratic discriminant analysis) can improve classification performance, but only marginally. Note that we know a priori that equal variance is unlikely as neurons exhibit Poisson spiking statistics (the variance is proportional to the mean); nonetheless, the regularization through a pooled variance estimate still allows improved generalization and thus is the lesser of two evils. Anyhow, we have updated the Materials and methods for a simplified and better justified.

Materials and methods :

“Classification was performed using linear discriminant analysis with the following parameter choices: one, only the mean firing rates differ for unit activity in response to each touch location (covariance of the normal distributions are the same for each condition); and, two, firing rates for each unit are independent (covariance of the normal distribution is diagonal). These choices do not reflect assumptions about the behavior of neurons, but instead, were found to improve cross-validation prediction accuracy on preliminary data. In our experiments, we acquired 10 repetitions per touch location, generally not enough data to robustly estimate the covariance matrix that describes the conditional dependence of the neural behavior on the stimulus. In choosing equal covariance, we are able to pool data across touch locations, achieving a more generalizable approximation of the neural response as verified by cross-validation.”

13) We computed the cross-validated coefficient of determination(R2 within) to measure how well a neuron's firing rate could be explained by the responses to the sensory fields.This needs a better description, and I may be missing the point entirely. I assume it is an analysis of mean firing rate (which should be stated explicitly) and that it uses something like the indicator variable of the linear analysis of individual neuron tuning above. In this case is this is a logistic regression? As it is computed for each side independently, it would appear that there are only four bits to describe the firing of any given neuron. This would seem to be a pretty impoverished statistic, even if the statistical model is accurate.

You are correct in saying that the model, a function built on indicator variables, is a description of the mean response to each tactile field, with the mean being captured by the continuously valued β coefficient. In comparing across body sides, we are asking whether the mean response to each tactile field computed from one side of the body can predict the single trial responses to each tactile field on the other side of the body. A few clarifications: while the indicator variable is composed of zeros and ones, the β values (means) that scale the indicator variable in the linear model are continuous values. Thus, it is not 4 bits, but instead, 4 continuous values. Further, if we had simply computed the mean values for each side, and then correlated the resulting sets of 4 values, we would not expect especially meaningful results as correlations computed with such a small number of values can by chance cover the whole spectrum of correlations. Instead, we are seeing how well the trial-to-trial responses of one side is explained by the mean responses of the other side. This is summarized as the R2 value. We have revised the description of the linear model for greater clarity, updated the description of the approach, and included a new schematic figure (Figure 3—figure supplement 4) to motivate the approach. This figure along with its caption was shown in response to major point 6.

The revisions to the relevant text on the linear model can be found in the response to major point 4.

And clarification of the across body-side regression can be found in the response to major point 6.

14) The purpose of computing a specificity index was to quantify the degree to which a neuron was tuned to represent information pertaining to one side of the body over the other.This is all pretty hard to follow. The R2 metric itself is a bit mysterious, as noted above. Within and across R2 is fairly straightforward, but adds to the complexity, as does SI, which makes comparisons of three different combinations of these measures across sides. Aside from R2 itself, the math is pretty transparent. However, a better high-level description of what insight all the different combinations provide would help to justify using them all. As is, there is no discussion and virtually no description of the difference across these three scatter plot. The critical point apparently, is that, "nearly all recorded PC-IP neurons demonstrate bilateral coding". There should be much a more direct way to make this point.

We have updated the text and streamlined the figure to make this section a little more digestible. These changes can be found in response to major point 6.

15) Computing response latency via RF discrimination is rather indirect and assumes that there is significant classification in the first place. I suspect it will add at least some delay beyond more typical tests. Why not a far simpler and more direct test of means in the same sliding window? Alternatively, a change point analysis?

The original sliding window classification analysis included three conditions: touch to the shoulder, cheek, and hand. The inclusion of the hand, a condition resulting in no significant modulation of the population, should allow for detection of the touch response, even in the absence of RF modulation. However, following your suggestion, we reanalyzed the latency data and have updated our method as outlined below. In brief, we now use a piece wise linear model on the first principal component (1PC) to detect the time the signal rises above the baseline response. A bootstrap procedure is used to find the quartile range of these times. Interestingly, the latency estimates are clearly earlier then what we got for the classification approach. We believe this is likely related to discussion above about Mahalanobis distance versus classification: namely, changes in the underlying neural behavior are only detected once the neural responses cross a decision line which likely results in delays detecting changes in neural behavior.

Here we opted for the 1PC as it is generally recommended when trying to capture the basic properties of the population response (51). Consistent with this, as shown in Author response image 2, the mean across the population (left) is quite a bit noisier then the first PC (right).

Another nice property of the first PC, is that save for averaging responses in 2ms bins, we did not impose any additional smoothing.We have updated the manuscript to include the new approach. The text from the Results and Materials and methods are copied below for convenience.

Results:

“We measured latency as the time at which the response of the neural population rose above the pre-stimulus baseline activity (Figure 6). The neural population response was quantified as the first principal component computed from principal component analysis (PCA) of the activity of all neurons (51, 52). The first principal component was then fit with a piece-wise linear function and latency was computed as the time the linear function crossed the baseline pre-stimulus response. Response latency was short for both body sides and was slightly shorter for contralateral (right) receptive fields (50 ms) than for ipsilateral (left) receptive fields (54 ms) although this difference was not statistically significant (Permutation shuffle test, *p*>0.05). Figure 6A shows the time course of the first principal component relative to time of contact of the touch probe (stepped window; 2 ms window size, stepped at 2 ms, no smoothing) along with the piece-wise linear fit (dashed line). A bootstrap procedure was used to find the inter-quartile range of latency estimates (Figure 6B)”

Materials and methods:

“We quantified the neural response latency to touch stimuli at the level of the neural population. […] We used a rank test to compare the true difference in latency estimates against an empirical null distribution of differences in latency estimates generated by shuffling labels and repeating the comparison 2000 times.”

[Editors' note: further revisions were suggested prior to acceptance, as described below.]

The reviewers were happy with the vast majority of the extensive revisions that have been implemented relative to the original submission, but felt that the presentation of the results and their discussion was inadequate. While we recognise the enormous amount of work that was poured into the revisions, there are still a few issues, originating from the revisions, that the reviewers felt require further consideration. But beyond these specific considerations that are elaborated below, we all agreed that the paper, at its present form, is too dense with results/interpretation, making it very difficult to read and evaluate. As one of the reviewers summarised:"My main high-level concern was the, "extensive and overlapping analyses" that made it difficult to follow and to find a clear takeaway message from the paper. I suggested eliminating a number of figures and clarifying the remainder to improve the impact of the paper. Although a few small panels have disappeared, most of the duplication remains and there are now nine, fairly complex figures (had been eight). It's hard to judge how much longer the text got because the figures are no longer embedded, but that has certainly also increased. The paper has not improved as a consequence".There is a strong need for a “deep clean” (or more precisely deep edit) of the paper. As highlighted in our original review, there's a lot of overlap between analyses/findings that will not be of interest to the average reader. We believe the key innovation of the study – the imagery results – could be presented a lot more concisely with a more focused discussion of the findings and the limitations of the suggested interpretation. Much of the details of the tactile RF properties is of secondary interest and as such should be moved to the supplementary section. I leave the decision of how to achieve this leaner and more focused version to the authors, but please consider the average reader (and their very limited time and attention span) when making the edits. As a rule of thumb, the reviewers estimated you will be likely be able to remove ~50% of the figure panels and 33% of the main text without weakening the key findings.

We understand the concern and have substantially reduced the amount of text and figures. We have cut the early portion of the text that discusses actual touch in half (from ~1800 words to ~850 words.) In addition we have cut the number of main figures in this section from 5 down to 2. This reduction was accomplished by a combination of removing figure panels (e.g. former panel 1B, 3B, 5C, 5E, portions of 5D and F were removed; we also removed 7C, 9B, and 9C) as well as pushing figures to the supplement (e.g. 3C, 4, and remaining parts of 5). This takes the main figures in the actual touch section from 18 panels down to 5 panels. This new presentation gets the key points across, e.g. neurons have spatially structured tactile receptive fields that are activated at short-latency consistent with processing of tactile sensations, in a more succinct manner.

This reorganization does leave a number of supplementary figures. However, the text is structured such that these figures are truly supplements, and can be skimmed or skipped without compromising a basic understanding of the primary message. We considered removing additional panels. However, 1) to our knowledge this is the first report of single unit responses in human PPC to tactile stimuli, and we think an (admittedly smaller) group of interested readers will find these analyses essential and 2) we have several upcoming papers what will cite these basic touch responses.

1) Perhaps the most exciting innovation of the study relates to the neural responses related to the imagery conditions. Yet, we know from many previous studies in humans that cognitive processes…While the Introduction has been much improved, the Discussion still mostly disregards the alternative cognitive processes that are likely to drive the present findings (prediction, attention). The authors seem to downplays the impact of attention considerably. For example, they state:"Most studies of pre-stimulus attention report that any modulation of baseline activity is modest at best."But especially when considering the modest effects of imagery, this statement is misleading. The Roland (1981) study that we provided, for example, show a 25% increase in rCBF when subjects attended the index finger without being stimulated. And this increase was spatially specific as it shifted to the lip area when attending to the upper lip instead of the finger. Although the other 3 attention references were looking at the effects of attention while sensory stimulation was present, they still seem relevant to the discussion and show what I'd consider to be greater than modest effects of attention. For example, the Puckett (2017) reference shows clear digit-specific modulation when attending to the different fingertips. Note that despite stimulation being present during this condition, the stimulation was constant (allowing the phasic attention-related signal to be somewhat isolated from the sensory signal). The amplitude of the phasic attention signal was similar to that elicited by phasic stimulation alone (i.e., without endogenous attention).There needs to be a more serious consideration that the effects attributed here to imagery are in fact modulated by (or even driven by) related cognitive processes, such as prediction and attention, which are not specifically linked to the auditory cue used in the present study.

We agree that the precise neural correlate of the activity is unclear. As the reviewer mentioned, imagery is just one possibility, and the manuscript does not state that the responses are the neural correlate of imagery.

For example, the Abstract states:

“Our results are the first neuron level evidence of touch encoding in human PPC and its cognitive engagement during tactile imagery, which may reflect semantic processing, attention, sensory anticipation, or imagined touch.”

Introducing the relevant section of the Discussion we write:

“While our task identifies dynamic engagement of multiple cognitive processes during tactile imagery, it is inadequate to precisely define the cognitive correlates of the observed neural activity. A number of cognitive processes may be engaged during the tactile imagery task including preparation for and/or execution of imagery, engagement of an internal model of the body, semantic processing of the auditory cue, allocation of attention to the cued body location or nature of the upcoming stimulus, and/or sensory memory for the corresponding actual sensation applied by the experimenter.“

That said, there are a couple of places where the distinction between the task that is used to evoke activity and the underlying cognitive process which is encoded could be made even more explicit. We have updated the text to more clearly specify that neural activity results from a “tactile imagery task”, e.g. updating the Abstract:

“We recorded neurons within the PPC of a human clinical trial participant during actual touch presentation and during a tactile imagery task. Neurons encoded actual touch at short latency with bilateral receptive fields, organized by body part, and covered all tested regions. The tactile imagery task evoked body part specific responses that shared a neural substrate with actual touch. Our results are the first neuron level evidence of touch encoding in human PPC and its cognitive engagement during a tactile imagery task, which may reflect semantic processing, attention, sensory anticipation, or imagined touch.”

We have also updated the portion of the Discussion that explicitly discusses a possible role for attention (We noticed that the references for neuroimaging were misaligned to our purpose in the revision. We had intended to use the citations the reviewer provided (here shown as 1-4). Thank you for references, and we have made the correction.):

“Hearing an auditory cue can direct the study participant’s attention to the cued body part. […] If our results are interpreted within the framework of attention, our current findings are inconsistent with a simple gain-like mechanism for attention, but instead suggest a richer mechanism by which information is selectively enhanced for further processing (7).”

This update is similar to the original, but I think the text is a fair accounting, consistent with our reading of the literature. First, we state that attention may be engaged by the task and that attention allocated to the body has been shown to effect neural responses. We are not questioning whether attention alters somatosensory processing. However, critically, this is not the question at hand. What is relevant to the current paper is whether attention generates condition specific changes in single unit neural firing, in the absence of any stimulus. The statement “Most studies of pre-stimulus attention report modest modulation of baseline neural activity“ is, to the best of our knowledge, an accurate statement of reported effects of pre-stimulus attention on measures of single unit activity. This is not a dismissal of attention, it just reflects the majority of studies that have looked at single unit spiking during the pre-stimulus period. In the revised text we go further than I believe necessary by stating that the inability to find a positive result may have been a failure to use sensitive analysis as suggested by the Snyder et al. paper. We then cite the one paper that shows relatively substantive pre-stimulus effects of attention in the firing of single units and state our results are compatible with the attention findings. We go on to interpret our results within the framework of attention. Again, I think the text is a fair accounting, consistent with our reading of the literature.

The reviewer provided references to bolster the case for attention. Three of the references measure attention effects on stimulus responses. We unequivocally state that “Attention to a stimulated body part has been shown to enhance sensory processing in human neuroimaging (1-4).” We are not questioning whether attention alters sensory processing and have included the recommended references to support a role for attention in the somatosensory system. As we state above however, it is not clear what this would say about the effect of prestimulus attention on spiking activity.

The reviewer also provides a reference to Roland 1981, which measures a 25% change in rCBF to attended body parts absent a stimulus. Can these changes be clearly attributed to attention-based modulation of single unit firing? To this point, Roland states “The most natural explanation for the rCBF and metabolic increase in the sensory finger region is that they mainly result from the local sum of many EPSPs … [and] a part of the metabolic increase in the sensory finger area may also be due to IPSPs.” (see Discussion). Roland thinks the bulk of the signal is dendritic processing, not a result of action potentials. Please understand that we are not using Roland’s words as a general critique of the ability of neuroimaging methods to measure single unit firing; we know that while they can dissociate, they generally correlate. That said, it is especially something like attention, defined in its modulatory capacity, that could give rise to dendritic computations in the absence of explicit spiking during the pre-stimulus period. Further, looking closer at Roland’s methodology, it is not clear that the rCBF results can clearly be attributed to attention: In Roland’s study, rCBF measurements were made absent an overt stimulus. However, only data in which the subject reported having experienced sensations were included in analysis. These “false alarms or reports of stimuli though none were applied … was considered a sign of high focal attention” otherwise “If no false alarms were reported the test was repeated.”. In this way “high focal attention” is inextricably tied to what might reasonably called imagined sensations, amongst other related cognitive variables. As with our paper, the highly correlated nature of many cognitive variables makes definitive interpretation difficult.

Taken together, I think that we have handled the possible interpretation of cognitive activity in a responsible manner.

2) Figure 4: Several panels would be more effective.This figure has changed quite lot, addressing my cosmetic concerns. However, I do not understand this statistical test: If no comparison was significant (FDR corrected), the unit was classified as "single peak." If at least one of the comparisons was statistically significant, it was characterized as "multi-peak." I must be missing something fundamental. I took this to be a test of differences of the responses to the different body parts, with respect to the peak response. How is this a single peak? No differences sounds like a flat line. By this definition a neuron with no response whatsoever would be "single peak". Likewise, the multi-peak definition is a puzzle.

From our results: “In brief, for each neuron, we found the location of maximal response and asked whether we could find a second local maxima that rose significantly above the neighboring values. If no significant second local maxima was found, the neuron was categorized as single peak, otherwise, the neuron was categorized as multi-peak.”

We have updated the Materials and methods description for improved clarity, copied below for convenience:

“We found that many neurons responded to touch to multiple body locations. We wished to further characterize the receptive field structure to determine whether neurons were characterized by single-peaked broad receptive fields or discontinuous receptive fields with multiple peaks. […] The unit was then classified as multi-peak. If no second local maxima was found the unit was classified as single-peak. ”

3) What is added by including both classification and Mahalanobis distance?"Mahalanobis distance that provides a sensitive measure of change which is masked by the discretization process of classification "I don't think the discrete nature of the classifier output is really the biggest issue. By averaging across many instances, it essentially becomes continuous, as in this figure. There are also classifier-related metrics that are by their nature, continuous. The nature of the distance measure they are making is likely more important. In this case, I do not understand why the classification rises only slightly at t=0 as the M distance increases sharply. Subsequently, between 1 and 2.5 s, classifier success drops back below its original level even as distance is stable. The two measures really don't seem to be concordant. What is going on here? I think this concern is not unrelated to my next comment about Figure 8C (now 9C).

There is a question generated from another question. It may be that only the last portion of this response is needed, but in the interest of clarity, I’ll introduce the topic more generally.

Neural state-space is encoded as discrete values when using classification. Consider the schematic in Author response image 3 in which the neural state can either be described using the continuous variables of firing rate, or the discrete variables of class label (A or B):

**Author response image 3. sa2fig3:** 

In this example, we see how two transitions (shown as vectors) through neural state space of comparable Mahalanobis distance are dramatically different in terms of classifier output (e.g. the movement started in class B and ended in class B, there is no indication that any movement occurred at all). From this example, it should be clear that Mahalanobis distance is a more sensitive measure of change. At the same time, it can also be significant whether or not the movement crosses significant boundaries. The classification result shows that although there is movement in neural state-space, the transition still leaves the neural state within the same basic class boundaries.Regarding how movement in state-space can give rise to the behavior shown, consider the following in Author response image 4:

**Author response image 4. sa2fig4:** 

Initially, the neural state moves further from the boundary (improving accuracy[Note that I am showing “mean” trajectories. Single trial results would be quite a bit noisier, accounting for trial-to-trial variability in accuracy]), while increasing Mahalanobis distance from the starting location. Towards the end of the movement, the position in state-space maintains a consistent distance from the starting point while drifting to the boundary. This would lead to a steady-state distance, but dropping accuracy. Other geometries, factoring in mean and variance, could account for the results. However, given that the changes in accuracy are on the order of a couple percentage points, it is not clear that a detailed analysis would be of interest to a general audience.

4) Figure 8C: Despite my best efforts, I have no idea…"This asymmetry is likely a consequence of the analysis technique and may not be of physiological significance.”I agree with the statement, but not its sentiment. Perhaps I'm missing something, but the fact that a single classifier can distinguish between rest state and two very different activation states does nothing to suggest those two states are a general representation of an input. The classification failure in the opposite direction only reinforces that. Presumably, classifying imagined and actual touch would be trivial, at a much higher level of success than rest and imagined touch, suggesting that they are in fact, rather different, even by this metric. If the authors wish to make the claim that their results show more than grossly common receptive fields bilaterally and across modes (which is not an uninteresting finding) they would do well to adopt tools more appropriate for it, like those that have been used by the groups of Shenoy, Churchland, Miller, Kaufman, and others: Canonical correlations, principal angles, subspace overlap.

I believe there is a misunderstanding here. Perhaps it is best to take a step back.

Our objective is to test the hypothesis that actual and cognitive representations of touch share a neural substrate at the population level in PPC. We address this by asking whether the neural population representations of the stimulated body part are similar across these two contexts. There are a number of metrics that can be used to measure similarity. In the draft, we use cross-classification accuracy as the metric of similarity. This approach creates a low-dimensional discretized representation of neural space (the decision space defined by the classifier) based on the conditions of one context (e.g. imagery) and asks whether trials from the alternate context map to corresponding regions of neural space. This is a powerful notion of similarity; it says that a decision maker (e.g. the classifier) interprets population level responses across contexts in a similar way.

Cross-classification is not the only way of computing a meaningful measure of similarity. The reviewer suggests a number of methods, with the implication that these alternate metrics would reveal something more fundamental about the relationship between the two contexts. While I agree that these proposed methods would measure something different, I disagree that they can reveal something more fundamental.

Consider principal angles. In this approach, we would compute the latent manifold independently for each context using, e.g. PCA. We would then compute the principal angles between these manifolds, thereby quantifying the degree to which population responses in the two contexts live in the same manifold. On the surface, this sounds interesting, and, no doubt, has its place. However, for our purposes, I would argue that it is less stringent and less revealing than something like cross-classification. This is because overlapping manifolds does not imply similar neural encoding. In fact, manifolds can be perfectly aligned in the potentially uninteresting case that the only correspondence between the two contexts is that the same set of neurons are modulated, but in completely unrelated ways. To illustrate, I created a toy problem (full matlab code below) where the two contexts are characterized by independent realizations of a multivariate normal distribution (e.g. X=randn(500,50); Y=randn(500,50)). I then added a multiplier on the first 10 dimensions. As shown in Author response image 5 (left panel) there is no covariance between the two contexts. Due to the multiplier term, there is a ten-dimensional manifold that explains >95% of the variance (Author response image 5 : middle column). Because the multiplier was applied to the same columns of the two contexts (X and Y), we see that the first 10 principal angles are close to 0 (Author response image 5 : right column), indicating that the two manifolds are very similar. As this example illustrates, near perfect alignment of manifolds might say nothing more than that the same neurons modulate in both contexts. This has its place, and may even be necessary to find any meaningful relationship when eg. Using unlabeled data or when temporal considerations prevent a meaningful alignment of the data. However, in our case, we wanted a metric that is able to say whether the same neurons are modulating in similar ways based on task conditions, not just that the same neurons are modulating. Cross-classification gives us this, principal angles do not.

Matlab code:

X=randn(500,50); Y=randn(500,50);

idx=1:10;

X(:,idx)=X(:,idx)*10; Y(:,idx)=Y(:,idx)*10;

[coeffX,scoreX,latentX,tsquaredX,explainedX,muX] = pca(X);

[coeffY,scoreY,latentY,tsquaredY,explainedY,muY] = pca(Y);

figure; subplot(1,3,1); hold on

imagesc(corr(X,Y),[-1 1]); colormap('jet');colorbar

axis image; title('Correlation')

xlabel('Context 1'); ylabel('Context 2')

subplot(1,3,2); hold on

plot(explainedX,'r.-','markersize',10)

plot(explainedY,'g.-','markersize',10)

ylabel('Var Explained'); legend({'Context 1','Context 2'})

theta=subspacea(coeffX(:,1:12),coeffY(:,1:12));

subplot(1,3,3);

plot(theta'*180/pi,'.-','markersize',10)

title('Principal Angle')

ylabel('Angle (deg)')

**Author response image 5. sa2fig5:** 

Canonical correlation analysis (CCA) is also proposed. Again, CCA has its place but I do not think it is correct to say that it would reveal something more interesting/fundamental. Consider the following example. Author response image 6 shows the response of two neurons in two contexts, color coded by condition. Cross-classification in this scenario would fail to show a similar relationship between labeled data between contexts. In contrast, CCA would show a very strong similarity (correlation values ~.8 in this simulated data). CCA is successful because it finds the linear projection that best equates these two scenarios (in this case, heuristically, by rotating Context 1 clockwise). In effect, CCA computes a measure of similarity, after an arbitrary linear transformation is applied to maximize similarity. This is fine for what it is (and sometimes necessary when e.g. observations from the two contexts might come from different sensors that have no natural correspondence) but I don’t think it would provide deeper insight given that we already show that there is correspondence prior to an optimized linear mapping.

**Author response image 6. sa2fig6:** 

None of this changes the fact that the asymmetries inherent in cross-classification can add complication without any clear benefit. In our case, I believe the asymmetry may not be especially interesting, thus adding complication without benefit. For this reason, we are substituting out cross-classification for cross-correlation. This maintains the benefits of direct tests of whether the condition specific patterns of activation are consistent across contexts without the inherent complications that come from different discretization of neural space depending on the dataset used for training. As can be seen, the basic pattern of results is essentially identical.Results:

“Cognitive processing during the cue-delay and imagery epochs of the tactile imagery task shares a neural substrate with that for actual touch

Finally, we look at how encoding patterns through time generalize between the tactile imagery and actual touch conditions. A dynamic correlation analysis was applied both within and across the imagery and actual touch condition types (Figure 6A). In brief, the neural activation pattern elicited to each body location was quantified as a vector, and these vectors were concatenated to form a population response matrix for each condition type and for each point in time. These vectors were then pair-wise correlated in a cross-validated manner so that the strength of correlation between conditions could be assessed relative to the strength of correlation within condition, and across time. We found that the neural population pattern that defined responses to actual touch was similar to population responses both during the cue-delay or the imagery phases of the imagery task (Figure 6A). This implies that cognitive processing prior to active imagery as well as during imagery share a neural substrate with actual touch. Sample neuronal responses that help to understand single unit and population behavior are shown in Figure 6B.”

5) Computing response latency via RF discrimination is rather indirect.The authors have adopted a more sensible and sensitive test of latency. I do not agree with this statement, however: "We believe this is likely related to discussion above about Mahalanobis distance versus classification: namely, changes in the underlying neural behavior are only detected once the neural responses cross a decision line which likely results in delays detecting changes in neural behavior." In what fundamental sense is classification significance different from a significant distance in M-space? It seems to me that the more likely explanation is simply that significant modulation precedes significant discrimination.

The classification involved discrimination between touch to three locations, the hand, shoulder and cheek. Touch to the insensate hand resulted in no neural modulation. For this reason, modulation of cheek and shoulder should enable discrimination against the unmodulated hand condition. Peak accuracy in this case would not be 100% but results should still be significant. The exact reason why classification failed to detect the time of modulation is a timely manner is a legitimate question, but moot given that we have removed the classification-based approach entirely.

However, to illustrate why the discretization problem (as defined in response 3) could contribute, consider the schematic in Author response image 7:

**Author response image 7. sa2fig7:** 

Depending on the rate of change of the neural activity, and exact location of the decision boundary, there could be a significant delay between latency as measured by time of modulation and time neural activity crosses a decision threshold.